# A Deep Reinforcement Learning Framework for Column Generation

**Cheng Chi**
University of Toronto

**Amine Mohamed Aboussalah**
New York University

**Elias B. Khalil**[*]
University of Toronto

**Juyoung Wang**
University of Toronto

**Zoha Sherkat-Masoumi**
University of Toronto

## Abstract

Column Generation (CG) is an iterative algorithm for solving linear programs (LPs) with an extremely large number of variables (columns). CG is the workhorse for tackling large-scale *integer* linear programs, which rely on CG to solve LP relaxations within a branch and price algorithm. Two canonical applications are the Cutting Stock Problem (CSP) and Vehicle Routing Problem with Time Windows (VRPTW). In VRPTW, for example, each binary variable represents the decision to include or exclude a *route*, of which there are exponentially many; CG incrementally grows the subset of columns being used, ultimately converging to an optimal solution. We propose `RLCG`, the first Reinforcement Learning (RL) approach for CG. Unlike typical column selection rules which myopically select a column based on local information at each iteration, we treat CG as a sequential decision-making problem: the column selected in a given iteration affects subsequent column selections. This perspective lends itself to a Deep Reinforcement Learning approach that uses Graph Neural Networks (GNNs) to represent the variable-constraint structure in the LP of interest. We perform an extensive set of experiments using the publicly available BPPLIB benchmark for CSP and Solomon benchmark for VRPTW. `RLCG` converges faster and reduces the number of CG iterations by 22.4% for CSP and 40.9% for VRPTW on average compared to a commonly used greedy policy. Our code is available at this link.

## 1 Introduction

Machine Learning (ML) for Mathematical Optimization (MO) is a growing field with a wide range of recent studies that enhance optimization algorithms by embedding ML in them to replace human-designed heuristics [Bengio et al., 2021]. Previous work has overwhelmingly focused on combinatorial or integer programming problems/algorithms for which all decision variables are explicitly given in advance. For example, in a knapsack problem, each binary variable represents the decision to include or exclude an item in the knapsack; even with hundreds of thousands of items, modern integer programming solvers or heuristics can assign values to some or all variables simultaneously without any memory issues. However, there are many optimization problems in which there are more decision variables than one could ever explicitly deal with. In VRPTW, for example, each binary variable represents the decision to include or exclude a *route*, of which there are exponentially many.

Column generation (CG) is an algorithm for solving linear programs (LPs) with a prohibitively large number of variables (columns) [Desaulniers et al., 2006]. It leverages the insight that in an optimal

---

[*]Corresponding author: `khalil@mie.utoronto.ca`.

36th Conference on Neural Information Processing Systems (NeurIPS 2022).

solution to LP, only a few variables will be active. In each iteration of CG, a column with negative reduced cost is added to a Restricted Master Problem (RMP, where "restricted" refers to the use of only a subset of columns) until no more columns with negative reduced cost are found. When that occurs, CG will have provably converged to an optimal solution to the LP. To solve an *integer* linear program, CG can be used as an LP relaxation solver within the so-called *branch and price* algorithm. In this work, we will focus on CG as an algorithm for solving large-scale linear programs; our conclusion section discusses the straightforward application of our method to the integer case.

A commonly used heuristic rule is to greedily select the column with the most negative reduced cost in each iteration. However, is this always the "optimal" column to add if one is interested in converging in as few iterations as possible? Could ML-guided column selection, based on the structure of the current instance being solved, make better selection decisions that speed up the convergence of CG? These are the questions we tackle in this work. We adopt the standard view that one is interested in solving problem instances that have the same mathematical formulation but whose *data* (objective function coefficients, constraint coefficients, num. of vehicles, etc.) differ and are drawn from the same distribution. In VRPTW, for example, the geographical locations of customers that must be served by the vehicles may vary, as do the corresponding service time windows.

To tackle these questions, we propose a ML framework to accelerate the convergence of the CG algorithm. In particular, we utilize RL to select columns for LPs with many variables, where the state of the CG algorithm and the structure of the LP instance are encoded using GNNs that operate on a bipartite graph representation of the variable-constraint interactions, similar to Gasse et al. [2019]. Q-learning is used to derive a column selection policy that minimizes the total number of iterations through an appropriately designed reward function. Our premise is that by directly optimizing for convergence speed using RL on similar LP instances from the same problem (e.g., CSP, VRPTW), one can outperform traditional heuristics such as the greedy rule. Our contributions can be summarized as follows:

1. **CG as a sequential task:** We bring forth the first integration of RL and CG which appropriately captures the sequential decision-making nature of CG algorithm. Our RL agent, RLCG, learns a Q-function which takes the future rewards (namely, total number of iterations) into consideration and therefore can make better column selections at each step. In contrast with prior work on RL for mathematical optimization, ours is, to our knowledge, the first to tackle the very widely applicable class of problems with exponentially many variables.

2. **Curricula for learning over a diverse set of instances:** In practice, instances of the same optimization problem may vary not only in their data, but also their size/complexity. For example, in the CSP, instances may vary in the number of rolls that must be cut. To enable efficient RL over a widely varying instance distribution, we show how a curriculum can be designed for a given dataset of instances with a minimal amount of domain knowledge.

3. **Evidence of substantial improvements over existing myopic heuristics:** We evaluate RLCG on the CSP, which is known as the representative problem in this domain [Ben Amor and Valerio de Carvalho, 2004, Desaulniers et al., 2006], and the VRPTW, another widely studied and applied problem. We compare RLCG with the commonly used greedy column selection strategy and an expensive, integer programming based one-step lookahead strategy described by Morabit et al. [2021]. Our algorithm converges faster than both of these in terms of the number of iterations and total time. Our results show the value of considering CG as a sequential decision-making problem and optimizing the entire solving trajectory through RL.

## 2   Related Work

Work on the use of RL to guide iterative algorithms can be traced back to Zhang and Dietterich [1995], who used RL for a scheduling problem. More recently, Dai et al. [2017] and Bello et al. [2016] proposed deep RL for constructing solutions to graph optimization problems; the survey by Mazyavkina et al. [2021] summarizes subsequent advances in this space. Dai et al. [2017] were the first to use GNNs in this setting, a line of work that has also grown substantially (e.g., Cappart et al. [2021]) including in integer programming [Gasse et al., 2019]. Relatedly, Tang et al. [2020]

apply RL to the cutting plane algorithm for integer linear programming, where a policy that selects "good" Gomory cuts is derived using evolutionary strategies. Within the framework of Constraint Programming, Cappart et al. [2020] present a deep RL approach for branching variable selection within an exact algorithm. Nonetheless, to our knowledge, the sequential decision-making perspective has not been leveraged in the context of column generation or integer/linear programming with many variables, a setting which is relevant to many practical applications such as resource allocation (e.g., CSP), routing problems (e.g., VRPTW), and airline crew scheduling [Barnhart et al., 2003].

The closest work to ours is that of Morabit et al. [2021]. They formulate the column selection in each CG iteration as a column classification task where the label of each column (select or not) is given by an "expert". This "expert" performs a one-step lookahead to identify the column which maximally improves the LP value of the Restricted Master Problem (RMP) of the next iteration. This is done with an extremely time-consuming mixed-integer linear program (MILP). The RMPs are encoded using bipartite graphs with columns nodes ($v$) and constraint nodes ($c$), where an edge between $v$ and $c$ in the graph indicates the contribution of a column $v$ to a constraint $c$. Each node of the graph is then annotated with additional useful information for column selection stored as node features. Morabit et al. [2021] then use a GNN to imitate the node selection of the expert using supervision, similarly to what was done for branching by Gasse et al. [2019].

In contrast, our work treats the column generation as a sequential decision-making problem and utilizes RL to select a column at each iteration of CG. Our GNN acts as a Q-function approximator that maximizes the total future expected reward. As such, our work focuses on directly reducing the total number of CG iterations, whereas Morabit et al. [2021] derive a classifier that does not consider the interdependencies across iterations, treating them as independent. One approach to accelerate CG further is to add multiple columns per CG iteration Desaulniers et al. [1999]; we discuss this extension in the Conclusion.

## 3 Preliminaries: Column Generation

We will use the canonical Cutting Stock Problem (CSP) to describe the CG method as is typically done in textbooks on the topic [Desaulniers et al., 2006]. CSP is a general resource allocation problem where the objective is to subdivide a given quantity of a resource into a number of predetermined allocations to meet certain demands so that the total usage of the resources (e.g., total number of size-fixed paper rolls) is minimized. Such minimization is achieved by finding a set of optimal divisions of each resource, or in other words, using a set of optimal cutting patterns to divide resources. Due to the CSP's combinatorial nature and its exponentially large set of possible patterns (variables), CG is used to solve the LP relaxation of CSP iteratively without explicitly enumerating all possible patterns.

The CSP formulation and column generation algorithm we use is a common modification of Gilmore and Gomory [1961]. The set of all feasible patterns $\mathcal{P}$ that can be cut from a roll is defined as:

$$\mathcal{P} = \left\{ x_k \in \mathbb{N}^n : \sum_{i=1}^{n} a_i x_{ik} \leq L, x_{ik} \geq 0 \quad \forall i \in \{1, 2, \ldots, n\}, \forall k \in \{1, 2, \ldots, |\mathcal{P}|\} \right\}.$$

where each pattern $p \in \mathcal{P}$ is represented using a vector $x_k \in \mathbb{N}^n$. With $a_i$ being a possible cut length from a roll with length $L$, each element of $x_k$ specifies how many such cuts with length $a_i$ are included in pattern $p$. For instance, assume the length of the resource roll $L$ is 4m, so all possible $a_i$s are 1m, 2m, 3m and 4m, and then one possible cutting pattern $p$ is represented by $x_k = (0, 2, 0, 0)$. Let $\lambda_p$ be the number of times pattern $p$ is used. The formulation with $\lambda_p$ being a decision variable is:

$$\min_{\lambda \in \mathbb{N}^{|\mathcal{P}|}} \left\{ \sum_{p \in \mathcal{P}} \lambda_p : \sum_{p \in \mathcal{P}} x_{ip} \lambda_p = d_i \ \forall i \in \{1, 2, \ldots, n\} \right\},$$

where the objective function minimizes the total number of patterns used, which is equivalent to minimizing the number of rolls used. The constraints ensure demand is met, while enforcing the integrality restriction on $\lambda_p$.

This problem has an extremely large number of decision variables as $\mathcal{P}$ is exponentially large. Therefore, the problem is decomposed into the Restricted Master Problem (RMP) and the Sub-Problem (SP). The RMP is obtained by relaxing the integrality restrictions on $\lambda_p$ with an initial set $\tilde{\mathcal{P}}$

where $\tilde{\mathcal{P}} \subset \mathcal{P}$. The RMP formulation of the cutting stock problem is defined as follows:

$$\min_{\lambda \in \mathbb{N}^{|\tilde{\mathcal{P}}|}} \left\{ \sum_{p \in \tilde{\mathcal{P}}} \lambda_p : \sum_{p \in \tilde{\mathcal{P}}} x_{ip} \lambda_p = d_i \ \forall i \in \{1, \cdots, n\}, \lambda_p \geq 0 \ \forall p \in \tilde{\mathcal{P}} \right\}.$$

The SP formulation of the provided cutting stock problem is defined as follows:

$$\min_{x \in \mathbb{N}^n} \left\{ \sum_{i=1}^{n} \pi_i x_i : \sum_{i=1}^{n} a_i x_i \leq L \right\},$$

where $\pi_i$ is the dual value associated with the demand constraints. The SP is used to generate a pattern $p$ represented by vector $x \in \mathbb{N}^n$ with the most negative reduced cost, then adding that $p$ to $\tilde{\mathcal{P}}$ in the next iteration. Below, we provide an overview of the column generation algorithm, noting that our method will intervene in Step 4 to make a potentially non-greedy column selection decision:

1. Solve the RMP to obtain $\lambda^\star$ and $\bar{\pi}$;
2. Update the SP objective function using $\bar{\pi}$;
3. Solve SP to obtain $x_i^\star$;
4. if $1 - \sum_{i=1}^{n} \bar{\pi}_i x_i^\star \leq 0$, add the column associated with $x_i^\star$ and return to step 1, else Stop.

## 4  The `RLCG` Framework

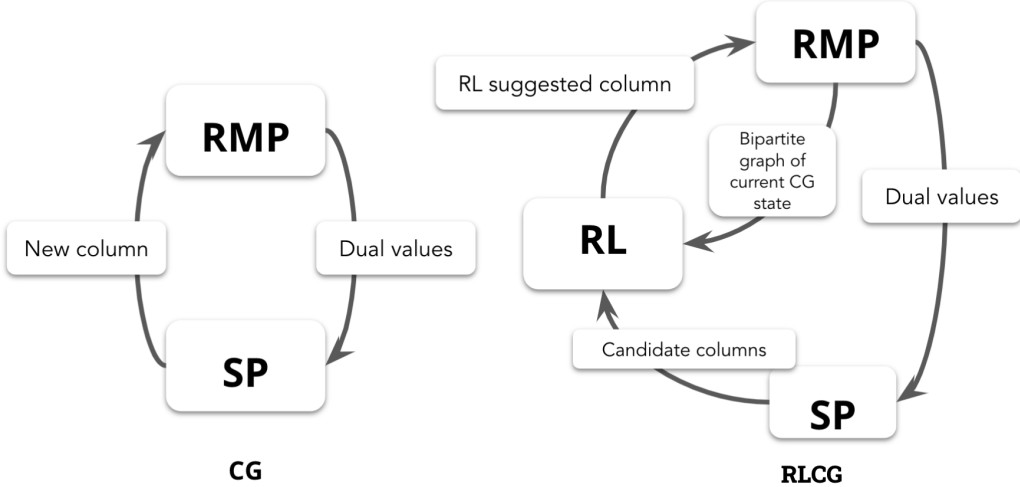

Figure 1: High-level comparison of standard Column Generation (CG) and `RLCG`.

At a high level, our RL-aided column selection strategy, `RLCG`, works as follows. We assume that the sub-problem (SP) is solved at each iteration and a set of near-optimal column candidates $\mathcal{G}$ is returned, which is a general feature of optimization solvers such as Gurobi. While the greedy CG algorithm, as described in Section 3, adds the single column with the most negative reduced cost from $\mathcal{G}$ to the RMP of the next iteration, `RLCG` selects columns from $\mathcal{G}$ according to the Q-function learned by the RL agent. The RL agent is fused within the CG loop and actively selects the column to be added to the next iteration using the information extracted from the current RMP and SP. An illustration comparing CG and `RLCG` is provided in Figure 1.

### 4.1  Formulating CG as MDP

We formulate CG as a Markov decision process (MDP). As is customary, we use $(\mathcal{S}, \mathcal{A}, \mathcal{T}, r, \gamma)$ to denote our MDP, where $\mathcal{S}$ is the state space, $\mathcal{A}$ the action space, $\mathcal{T} : \mathcal{S} \times \mathcal{S} \times \mathcal{A} \rightarrow [0, 1]$, $(s', s, a) \mapsto \mathbb{P}(s'|s, a)$ the transition function, $r : \mathcal{S} \times \mathcal{A} \times \mathcal{A} \rightarrow \mathbb{R}$ the reward function, and $\gamma \in (0, 1)$ the discount factor. We train the RL agent with experience replay [Mnih et al., 2015].

### 4.1.1 State space $\mathcal{S}$

The state represents the information that the agent has about the environment. In RLCG, the environment is the CG solution process corresponding to a given problem instance. As shown in Figure 1, the information passed to the RL agent is the bipartite graph of the current CG iteration from the RMP and the candidate columns from the SP. At each iteration, the RMP is an LP as shown in Section 3. As introduced in Gasse et al. [2019], such an LP is encoded using a bipartite graph with two node types: column nodes $\mathcal{V}$ and constraint nodes $\mathcal{C}$. An edge $(v, c)$ exists between a node $v \in \mathcal{V}$ and a node $c \in \mathcal{C}$ if column $v$ contributes to constraint $c$. An example bipartite graph representation of the state is shown in left of Figure 2 with column nodes shown on the left hand side, the constraint nodes on the right hand side, and the action nodes, which are the candidate columns returned from the SP, are shown in green (e.g., $v6, v7$).

To incorporate richer information about the current CG iteration, we include node features for both column nodes and constraint nodes (**vf** and **cf** next to the nodes). We designed 9 column node features and 2 constraint node features in our environment based on our previous experience with CG and inspiration from Morabit et al. [2021]. These node features are described in Appendix G.

As such, the state space $\mathcal{S}$ is the space of bipartite graphs representing all possible RMPs from the problem instances drawn from the distribution $\mathcal{D}$ with the node features given above. The bipartite graph shown in the left of Figure 2 is a particular state $s$ in $\mathcal{S}$. As states are bipartite graphs with node features, it is natural to use a Graph Neural Network (GNN) as the Q-function approximator in our DQN agent. This bipartite graph representation not only encodes variable and constraint information in the RMP, but also interactions between variables and constraints through its edges.

### 4.1.2 Actions, transition, and reward

**Action space $\mathcal{A}$.** As shown in Figure 2, the RL agent selects one column to add to the RMP for the next iteration from the candidate set $\mathcal{G}$ returned from current iteration SP. Therefore, the action space $\mathcal{A}$ contains all possible candidate columns that can be generated from SPs; for example, the green nodes $v6$ and $v7$ in Figure 2. As the action space is discrete and the state space is continuous, the GNN (Q-network) performs the operation of returning action values in the current state, i.e., $\hat{q}(s, a_1; w), \ldots, \hat{q}(s, a_m; w)$.

**Transition function $\mathcal{T}$.** Transitions are deterministic. After selecting an action from the current candidate set $\mathcal{G}$, the selected column enters the basis in the next RMP iteration. We then delete the action nodes that were not selected from the bipartite graph (current state), turn the selected action node into a column node, solve the new CG's RMP and SP, update all the features, and augment all the action nodes returned from the next SP iteration into the left-hand side of the graph, which results in a new bipartite graph state. Take the bipartite graph state shown in the left of Figure 2 as an example, and assume action $v6$ is selected at this iteration. The transition occurs as follows: $v6$ (in grey) becomes a column node, and candidate columns $v8$ and $v9$ returned by SP (in green) are added.

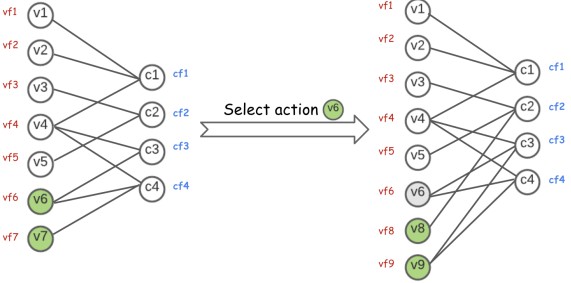

Figure 2: State transition: Two green action nodes are considered, one of them is selected, transitioning to a new state.

**Reward function $\mathcal{R}$.** The reward function consists of two components: (1) the change in the RMP objective value, where a bigger decrease in value is preferred; (2) a unit penalty for each additional iteration. Together, they incentivize the RL agent to converge faster. The reward at time step $t$ is defined as:

$$r_t = \alpha \cdot \left( \frac{\text{obj}_{t-1} - \text{obj}_t}{\text{obj}_0} \right) - 1, \tag{1}$$

where $\text{obj}_0$ is the objective value of the RMP in the first CG iteration and is used to normalize $(\text{obj}_{t-1} - \text{obj}_t)$ across instances of various sizes; $\alpha$ is a non-negative hyperparameter that weighs the normalized objective value change in the reward.

## 4.2 `RLCG` training and execution

Algorithm 1 shows how a trained `RLCG` agent is applied to solve a problem instance. The MDP components $(\mathcal{S}, \mathcal{A}, \mathcal{T}, r, \gamma)$ used in this section are defined in Section 4.1. Before starting the iterative optimization, the initialization steps 1–3 build the initial bipartite graph with computed node features and add an initial set of columns into basis (for instance, in CSP, we first add simple cutting patterns into the basis to initialize). Inside the while loop, a column is selected from the candidate set $\mathcal{G}$ based on the best Q-value computed by the RL agent. Then, the RMP and the SP are updated in the same way as the traditional CG algorithm. The MDP model corresponds to extracting the MDP components from the current updated RMP and SP, which are discussed in Section 4.1.1 and Section 4.1.2. Steps 7, 8 correspond to the deterministic state transition $\mathcal{T}$ described in Section 4.1.2, and $S_{t+1}$ is the resulting state due to action $a_t^*$.

---

**Algorithm 1:** `RLCG` (RL-aided column generation algorithm)

---

**Input:** Problem instance $p$ from distribution $\mathcal{D}$ & trained Q-function.
**Output:** Optimal solution
1  $t = 0$; $\text{RMP}_0 = \text{Initialize}(p)$
2  Solve $\text{RMP}_0$ to get dual values; Use dual values to construct $\text{SP}_0$.
3  $\langle S_0, A_0, T_0, R_0 \rangle = \text{MDP}(\text{RMP}_0, \text{SP}_0)$
4  **while** *CG algorithm has not converged* **do**
5  $\quad$ $a_t^* = \arg\max_{a_t \in A_t} Q(S_t, a_t) \quad \forall a_t \in A_t$ ;
6  $\quad$ Add variable $a_t^*$ to $\text{RMP}_t$ and get $\text{RMP}_{t+1}$
7  $\quad$ Solve $\text{RMP}_{t+1}$ to get dual values; Use dual values to build $\text{SP}_{t+1}$.
8  $\quad$ $\langle S_{t+1}, A_{t+1}, T_{t+1}, R_{t+1} \rangle = \text{MDP}(\text{RMP}_{t+1}, \text{SP}_{t+1})$
9  $\quad$ $t = t + 1$
10  **end**
11  Return optimal solutions from $\text{RMP}_t$, $\text{SP}_t$

---

**Training.** The DQN algorithm with experience replay is used [Mnih et al., 2015] with the typical mean squared loss between $Q(s_0, a)$ and $Q_{\text{target}}(s_0, a) \,\forall\, a$ (all green action nodes) in bipartite graph state $s_0$. $Q_{\text{target}}(s_0, a)$ is defined as $r + \gamma \cdot \max_a Q(s_1, a)$. A GNN is used as Q-function approximator. Training instances are sequenced based on a domain-specific curriculum described for each of CSP and VRPTW in Section 5.

# 5 Experimental Results

**Baseline strategies and evaluation metrics.** To assess the performance of `RLCG`, we consider two baseline methods: the greedy column selection strategy (most negative reduced cost) and the MILP expert column selection of Morabit et al. [2021] as described in Section 2. This expert provides an upper bound on the performance of the supervised learning approach of Morabit et al. [2021], as their ML model is approximately imitating the expert, i.e., it will never select better columns than the expert. We consider two standard evaluation metrics: (1) Number of iterations for CG to converge; (2) Time in seconds. For the latter, this includes GNN inference time and feature computations for `RLCG`. Our computing environment is described in Appendix Section C.

### 5.1 Cutting Stock Problem (CSP)

**Dataset.** We use BPPLIB Delorme et al. [2018], a widely used collection of benchmark instances for binary packing and cutting stock problems, which includes a number of instances proven to be difficult to solve [Delorme and Iori, 2020, Wei et al., 2020, Martinovic et al., 2020]. BPPLIB contains instances of different sizes with the roll length $n$ varying from 50 to 1000 and the number of orders $m$ varying from 20 to 500. Our training set has instances with $n = 50, 100, 200$. The remaining instances with $n = 50, 100, 200, 750$ are split into a validation set and a testing set, with hard instances $n = 750$ only appearing in the testing set, as it is very expensive to solve such large instances during training. The detailed statistics of these three sets are listed in Table 4 in Appendix E. Note that our test set is more challenging than that used in training, allowing us to assess the agent's generalization ability.

**Curriculum Design.** In the RL context, curriculum learning Narvekar et al. [2020] serves to sort the instances that a RL agent observes during the training process from easy to hard. In our experiments, we noticed that adopting the curriculum learning paradigm improves the learning of the RL agent and results in better column selection and faster convergence. We train our `RLCG` agent by feeding the instances in order of increasing difficulty. For CSP, instances are sorted based on their roll length $n$ and number of orders $m$ to build a training curriculum (Table 5 in Appendix C). The detailed comparison of training with and without the curriculum is deferred to Appendix A. In short, the former is crucial to convergence when the training set contains instances of varying difficulties. The overall training process

**Hyperparameter tuning.** We briefly describe the outcome of a large hyperparameter tuning effort which is described at length in Appendix B. We tune the main hyperparameters, $\alpha$ in the reward function (1), $\epsilon$ the exploration probability in DQN, $\gamma$ the discount factor, and the learning rate $lr$ in gradient descent. A grid with 81 possible configurations is explored by sampling 31 configurations, training the agent, and evaluating them on a validation set. The majority of the configurations resulted in agents that outperform the greedy strategy (see Appendix B Figure 10). The best configuration is: $\alpha = 300$, $\epsilon = 0.05$, $\gamma = 0.9$ and $lr = 0.001$. For all the experiments conducted below, we use this configuration, both for CSP and VRPTW.

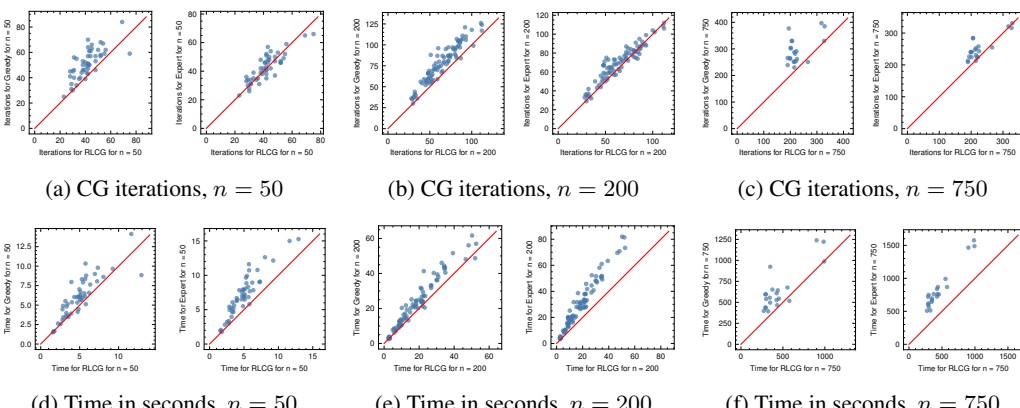

(a) CG iterations, $n = 50$     (b) CG iterations, $n = 200$     (c) CG iterations, $n = 750$

(d) Time in seconds, $n = 50$     (e) Time in seconds, $n = 200$     (f) Time in seconds, $n = 750$

Figure 3: **CSP:** Scatter plots of CG iterations (top row) and running time (bottom row) with `RLCG` on the horizontal axis and Greedy (Expert) on the vertical axis, respectively, in each of three pairs of sub-figures. Each point represents a test instance of a given size $n$. Points above the diagonal indicate that `RLCG` is faster than the competitor.

**Performance comparison.** Figure 3 shows pair-wise comparisons between `RLCG` and the two baselines in terms of the number of CG iterations (top row) and solving time (bottom row) on the testing instances. Each point in a plot corresponds to a test instance. Across all $n$ and for both baselines in Figure 3, the majority of the points are above the diagonal line, which indicates that `RLCG` outperforms the competing method w.r.t. the evaluation metric. Such a tendency becomes more pronounced as the CSP instances become larger (left to right). `RLCG` performs better than the greedy

column selection for all the roll lengths $n$ and the performance improvement becomes larger as $n$ grows. For CSP instances with $n = 50$ and $n = 200$, the MILP expert is able to maintain similar performance compared to RL. However, solving the MILP expert requires solving a MILP at each iteration with only one-step lookahead which is time consuming. For CSP instances with $n = 750$, RLCG begins to outperform the MILP expert due to its ability to take future rewards into consideration.

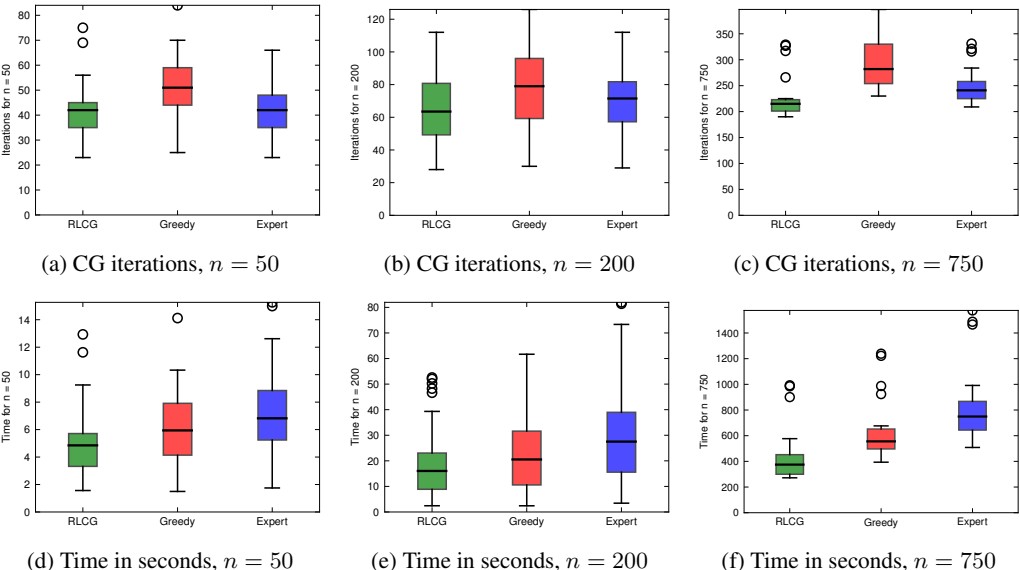

(a) CG iterations, $n = 50$      (b) CG iterations, $n = 200$      (c) CG iterations, $n = 750$

(d) Time in seconds, $n = 50$      (e) Time in seconds, $n = 200$      (f) Time in seconds, $n = 750$

Figure 4: **CSP:** Box-plots of CG iterations (top row) and running time (bottom row) for the proposed method, RL (green), Greedy (red), and Expert (blue). Each box represents the distribution of the iterations or running time for a given method and roll size $n$. Lower is better.

Notice that even though we did not train RLCG using CSP instances with $n = 750$, it was able to perform well on such instances, indicating strong generalization to harder problems. We also generate box plots to compare statistically the three methods shown in Figure 4. Detailed statistics can be found in Table 6. As shown by all the subplots in Figure 4, especially for large instances in the subplots (c) and (f), the proposed RLCG achieves statistically meaningful improvements on the CG convergence speed compared to both the greedy and the expert column selection methods.

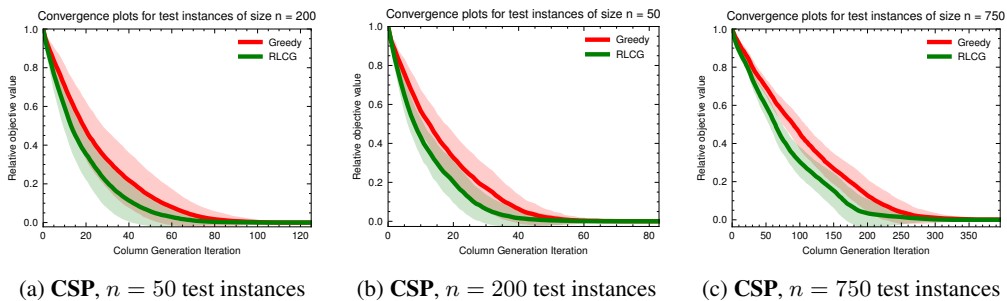

(a) **CSP**, $n = 50$ test instances      (b) **CSP**, $n = 200$ test instances      (c) **CSP**, $n = 750$ test instances

Figure 5: CG convergence plots for **CSP**. The solid curves are the mean of the objective values for all instances and the shaded area shows $\pm 1$ standard deviation.

**Convergence plots.** In Figure 5, we visualize the CG solving trajectories for all test instances for $n = 50, 200, 750$. We record the objective values of the RMP at each CG iteration for given method, then take the average over all instances. Note that we normalized the objective values to be in [0,1] before taking the average among instances. Since the MILP expert is by far the slowest, we restricted our attention to the two less expensive methods: RLCG and Greedy. Looking at Figure 7, it is clear

that not only does `RLCG` terminate in fewer iterations and less time, but it also dominates Greedy throughout the CG iterations. In other words, if one had to terminate CG earlier, `RLCG` would result in a better (smaller) objective value as compared to what Greedy would achieve.

## 5.2 Vehicle Routing Problem with Time Windows (VRPTW)

The VRPTW seeks a set of possible routes for a given number of vehicles to deliver goods to a group of geographically dispersed customers while minimizing the total travel costs. In the language of CG, each route is one column and there are exponentially many. Constraints of VRPTW include: vehicle capacity; a vehicle has to start from a depot and return to it; all customers should be served exactly once during their specified time windows. The detailed formulation of VRPTW is given by Cordeau et al. [2002]; implementation details are in the appendix.

**Dataset.** We use the well-known Solomon benchmark [Solomon, 1987]. This dataset contains six different problem types (C1, C2, R1, R2, RC1, RC2), each of which has 8–12 instances with 50 customers. "C" refers to customers which are geographically clustered, "R" to randomly placed customers, "RC" to a mixture. The "1" and "2" labels refer to narrow time windows/small vehicle capacity and large time windows/large vehicle capacity, respectively. The difficulty levels of these sets are in order of C, R, RC. There are 56 instances in total in Solomon's dataset, and from each original Solomon instance, we can generate smaller instances by considering only the first $n < 50$ customers.

We use instances from types C1, R1, RC1 for training. For the training set, we generated 80 smaller instances per type from the original Solomon's instances by only considering the first $n$ customers where $n$ is randomly sampled from 5–8, for a total of 240 training instances. For testing, two sets of instances are considered: (1) 60 small-size instances (number of customers $n$ within same range as training instances); (2) 37 large-size instances (number of customers $n$ from 15–30). All the test instances are either generated from held-out Solomon instances (e.g., sets C2, R2, RC2) or generated from some training instance but with a much different $n$.

**Curriculum Design.** As in CSP, we designed a curriculum for VRPTW that sequences instances in order of difficulty: C1, R1, then RC1. This order is based on the fact that clustered instances have more structure that enables "compact" routes for neighboring customers, whereas random instances require more complex routes. The mixed RC1 instances require reasoning about both types of customers simultaneously [Desrochers, 1992].

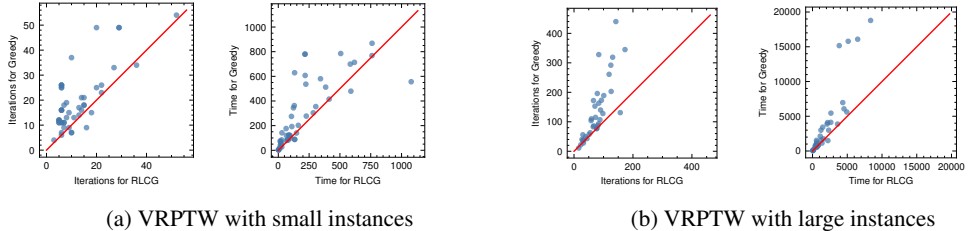

(a) VRPTW with small instances         (b) VRPTW with large instances

Figure 6: **VRPTW:** Scatter plots of CG iterations and running time with `RLCG`. Each point represents a test instance of a given size $n$. Points above the diagonal indicate that `RLCG` is faster than greedy.

**Performance comparison.** The hyperparemeters of the RL agent are chosen to be the same as the best set of hyperparameters found for CSP. Figure 6 shows similar trends to those seen for CSP: `RLCG` converges in fewer iterations and less time than Greedy on most instances.

This effect is more pronounced for the large VRPTW instances, a fact that can also be observed in the convergence plot of Figure 7 (b): `RLCG` converges in roughly 100 iterations compared to 300 iterations for Greedy, on average. Additional box plots are deferred to Appendix J. Note that we do not compare to the Expert here given that it is much too slow and practically intractable.

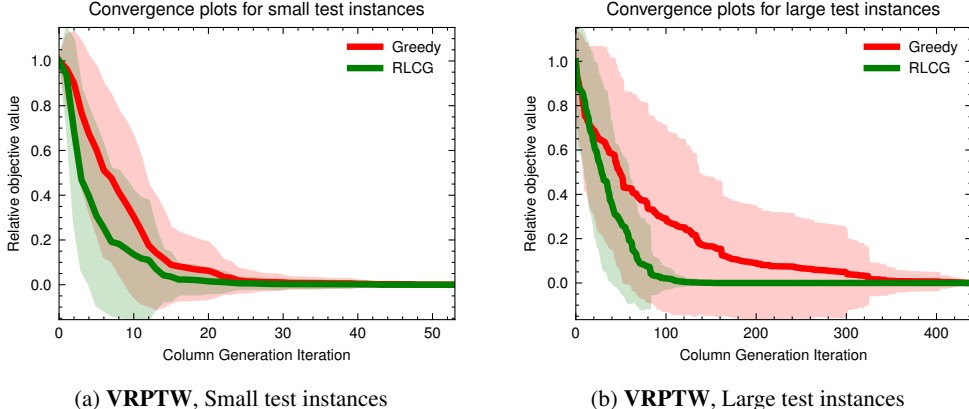

(a) **VRPTW**, Small test instances       (b) **VRPTW**, Large test instances

Figure 7: CG convergence plots for **VRPTW**. The solid curves are the mean of the objective values for all instances and the shaded area shows $\pm 1$ standard deviation.

## 6 Conclusions & Discussion of Limitations

RLCG shows superior performance and better convergence both in terms of the number of iterations and time compared to the greedy column selection and the MILP expert strategy. In addition, our curriculum learning enables the agent to generalize well when facing harder test instances. Our experiments on two important large-scale LP families show that there is value in modeling CG as a sequential decision-making problem: taking the future impact of adding a column into account helps convergence.

However, our current work is restricted to adding only one column per CG iteration. Adding multiple columns per iteration can speed up convergence. However, this makes the RL action space exponential and thus finding the action with the largest Q-value becomes a combinatorial optimization problem. Recent work by Delarue et al. [2020] addresses this problem, and thus our RL formulation can be expanded using the results of this paper to allow for multiple columns. Alternatively, policy gradient methods could be used instead of Q-learning. We believe this is an exciting next step but one that stretches beyond the limits of our paper, which is the first ever on RL for CG. A trained RLCG agent can also be embedded within the branch and price algorithm for solving the integer-constrained versions of CSP/VRPTW and invoked to solve each LP relaxation. The speed-ups demonstrated herein would transfer to that setting, assuming that an appropriate dataset of training and validation instances can be collected.

**Acknowledgments:** Khalil acknowledges support from the Scale AI Research Chair Program and an NSERC Discovery Grant.

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
