# A Curriculum learning

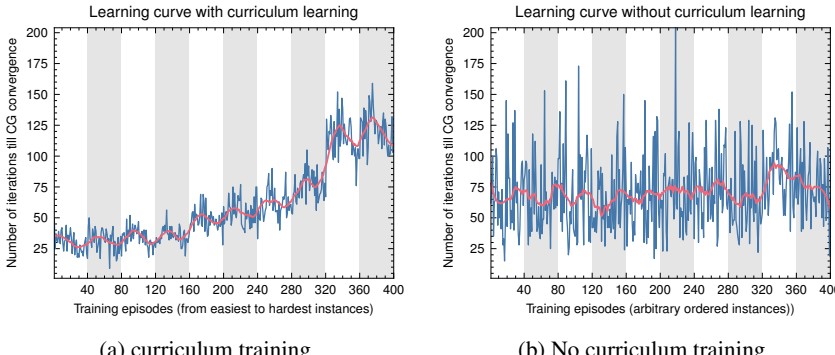

(a) curriculum training        (b) No curriculum training

Figure 8: (a) shows the training process with such curriculum and Figure (b) shows the training process without a curriculum. In both plots, the x axis is the training episode, and each episode is solving one instance (instances are ordered for (a) and randomized for (b)) while the y axis shows the total number of RL guided CG iterations until that particular instance is solved.

Figure 8 above compares the training trajectories between training with 400 CSP instances following the sequence provided in Table 5 and training with the same 400 CSP instances randomly ordered. In (a), as every 40 instances we increase the CSP difficulty. Although there is an upward trend in the training curve, however, within each instance difficulty setting (fixed n and m), there is a downward trend showing a sign of learning. In contrast, there is no clear sign of learning in (b). Therefore, for all the experiments shown in this paper, the RLCG model is trained using a curriculum.

We also visualize the training process of RLCG for CSP using a validation set with 30 instances. The validation set detail is in Appendix E. For every 20 training episodes, we stop the training process and validate the current models (with schedule training and without schedule training) with the validation set. The result is shown in Figure 9.

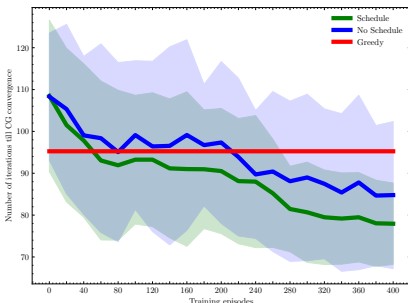

Figure 9: Green curve shows the mean of convergence iterations for the current RLCG model with schedule training solving all the validation instances, and plus minus one standard deviation is shown in shaded area. Blue curve shows validation result of model trained without schedule. The red horizontal line shows average convergence iterations for validation instances using greedy strategy

# B Hyperparameter tuning

Table 1 shows RL related parameters.

In Table 2, we provide GNN parameters that we used.

We conduct hyperparameters tuning and sensitivity analysis using a validation set over: the parameter $\alpha$ used in equation 1 to weight the change in the normalized objective value used in our reward function, the exploration parameter of the RL agent $\epsilon$, the discount factor $\gamma$, and the learning rate

Table 1: RL Parameters

| Parameter | Value |
|---|---|
| state normalization | features normalization is *MinMaxScalar* from sklearn |
| step penalty | For each iteration RLCG takes before the current CSP instance is solved, penalize each step by 1 in the reward design |
| reward | there are two settings $\alpha = 5$ and $\beta = 1$, $\alpha = 5$ and $\beta = 0$, step penalty is always 1 |
| action | solution pool 10 from Gurobi solver for SP, which means at each iteration our action space contains 10 columns. |

Table 2: GNN Parameters

| Parameter | Value |
|---|---|
| Optimizer | Adam |
| Network Structure | refer to code for details |
| Batch Size | 32 |

$lr$. All other hyperparameters and their values are listed in Table 1 and Table 2. The values we consider for each hyperparameter are the following: $\alpha \in \{0, 100, 300\}$, $\epsilon \in \{0.01, 0.05, 0.2\}$, $\gamma \in \{0.9, 0.95, 0.99\}$ and $lr \in \{0.01, \text{1e-3}, \text{3e-4}\}$. We choose the value for $\alpha \in \{0, 100, 300\}$ because when $\alpha = 0$, we place no weight on decreasing the objective value, otherwise, $\alpha = 100, 300$ will bring the normalized change in the objective values into similar scale as the step penalty. Therefore, the search space for hyperparameters is defined as the Cartesian product between all these sets of different hyperparameters possible values, which gives us 81 configurations, and we randomly select 31 configurations out of them. Then we train 31 RLCG models corresponding to selected 31 configurations, and we evaluate these models using our validation set. The validation metric or reward is defined as the ratio of the total number of iterations RLCG takes to solve each CSP instance divided by the total number of iterations greedy takes. For each model, we compute such ratio for all the instances in validation set, and we generate the following box plot shown in Figure 10 showing the validation metric for all the models. Among the 31 configurations we tested, the majority of the models were able to outperform greedy strategy on the instances in the validation set. The best configuration is model 3: $\alpha = 300$, $\epsilon = 0.05$, $\gamma = 0.9$ and $lr = 0.001$. For all the results reported in this paper, we use this configuration. This includes VRPTW, although no direct hyperparameter tuning has been done for this problem class. To assess the sensitivity of the RL training with respect to randomness such as the GNN initialization and the exploration in RL, we compare the average validation reward relative to greedy for the selected model across five random seeds. The average varies between 1.23 and 1.25, indicating little to no sensitivity.

In Table 3, we provide detailed configurations of each model index for our analysis of hyperparameters as well as their detailed validation results.

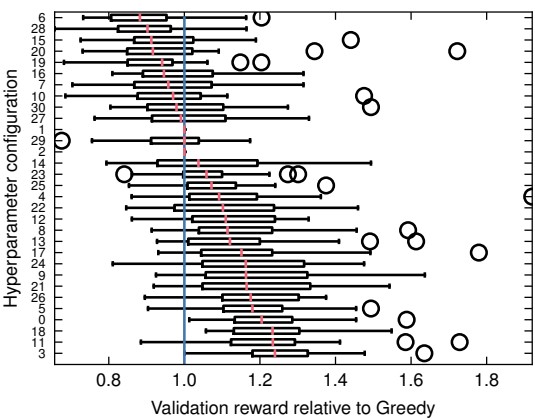

Figure 10: Hyperparameter sensitivity: the vertical axis list all the models we evaluated by its index. The detailed hyperparameter configurations each index refers to are listed in Appendix B Table 10. The horizontal axis shows the ratio of `RLCG` model solving iterations to greedy column selection solving iterations. The blue vertical line shows the ratio threshold indicating same performance as greedy. The best to worst models are ordered from bottom to top.

## C    Computing environment

To implement GNN, we use Tensorflow 2.7.0. To solve both RMP and SP optimization problems in CG, we use Gurobi 9.5.0. For the training using 400 instances schedule for CSP and 240 instances schedule for VRPTW, the training takes around 8-10 hours CPU time using the following CPU settings: Intel 2.30 Ghz, 2 CPU Cores, Intel (R) Xeon(R), Haswell CPU family.

## D    Graph neural network for bipartite graph

Graph Neural Network (GNN) has been successfully applied to many different machine learning tasks with graph structured data. GNN includes a message passing method where the features of each node for each node pass to other neighbouring nodes through learned transformations to generate aggregated information, and such information can be used for node classification, edge selection so on so forth. Due to the effectiveness of GNN to utilize graphical structure of the data, catch node-level dependencies, and has permutation invariant properties, GNN is an appropriate method for performing node(column) selection task in the present column generation problem.

For this study we are encoding the information for each iteration of CG using a bipartite graph $G = (E, V)$ as the state, where each column and constraint is represented by node $v \in V$ and there is an edge $e \in E$ between columns and constraints only if the column contributes to the constraint. Detailed encoding for state is discussed in Section 4.1.1. As we are using a bipartite graph, the GNN we use should be able to achieve convolution on a bipartite graph with two types of nodes (variable nodes and constraint nodes), and we utilize the similar bipartite GNN as in the study conducted by (Morabit et al. [2021]) with modification of the task it performs (from binary classification to Q value regression). Here we give a brief overview of how convolution or the features update is achieved in this bipartite GNN.

The features update is done in two phase: phase 1 updates the constraint features and phase 2 updates the column features. In phase 1, the constraint features in next iteration is obtained by applying a non-linear transformation of previous constraint node features and aggregate information of its neighbouring nodes, which are column nodes that are connected to this constraint node. This can be treated as information passing from variable nodes to constraint nodes. Similarly,the second phase can be seen as message passing from constraint nodes to variable nodes.

Once the column node features have been updated for several iterations for all the column nodes, these features are fed into a fully connected layer, which results in Q values for each column node.

| Model index | Hyperparameters config | iterations mean | iterations median | iterations std |
|---|---|---|---|---|
| Model 0 | (100, 0.2, 0.9, 0.0003) | 1.22 | 1.20 | 0.14 |
| Model 1 | (100, 0.05, 0.99, 0.01) | 1.00 | 1.00 | 0.00 |
| Model 2 | (100, 0.2, 0.9, 0.01) | 1.00 | 1.00 | 0.00 |
| Model 3 | (300, 0.05, 0.9, 0.001) | 1.25 | 1.24 | 0.13 |
| Model 4 | (300, 0.05, 0.99, 0.001) | 1.12 | 1.09 | 0.19 |
| Model 5 | (300, 0.05, 0.95, 0.001) | 1.18 | 1.18 | 0.14 |
| Model 6 | (0, 0.05, 0.9, 0.0003) | 0.91 | 0.88 | 0.13 |
| Model 7 | (0, 0.2, 0.95, 0.0003) | 0.98 | 0.96 | 0.14 |
| Model 8 | (100, 0.05, 0.9, 0.0003) | 1.14 | 1.11 | 0.16 |
| Model 9 | (100, 0.2, 0.9, 0.001) | 1.19 | 1.16 | 0.18 |
| Model 10 | (0, 0.05, 0.95, 0.0003) | 0.97 | 0.97 | 0.14 |
| Model 11 | (300, 0.01, 0.95, 0.001) | 1.22 | 1.23 | 0.16 |
| Model 12 | (300, 0.05, 0.9, 0.001) | 1.11 | 1.11 | 0.12 |
| Model 13 | (100, 0.05, 0.95, 0.0003) | 1.15 | 1.12 | 0.16 |
| Model 14 | (100, 0.2, 0.9, 0.001) | 1.07 | 1.04 | 0.17 |
| Model 15 | (100, 0.05, 0.9, 0.001) | 0.95 | 0.91 | 0.15 |
| Model 16 | (0, 0.2, 0.99, 0.001) | 0.99 | 0.95 | 0.13 |
| Model 17 | (300, 0.2, 0.95, 0.001) | 1.18 | 1.15 | 0.17 |
| Model 18 | (300, 0.2, 0.9, 0.0003) | 1.24 | 1.23 | 0.13 |
| Model 19 | (0, 0.05, 0.9, 0.0003) | 0.92 | 0.94 | 0.12 |
| Model 20 | (0, 0.2, 0.9, 0.001) | 0.95 | 0.92 | 0.19 |
| Model 21 | (300, 0.05, 0.99, 0.0003) | 1.18 | 1.16 | 0.16 |
| Model 22 | (300, 0.2, 0.99, 0.001) | 1.11 | 1.10 | 0.16 |
| Model 23 | (100, 0.2, 0.9, 0.01) | 1.06 | 1.06 | 0.10 |
| Model 24 | (300, 0.05, 0.9, 0.001) | 1.16 | 1.16 | 0.17 |
| Model 25 | (100, 0.2, 0.95, 0.001) | 1.07 | 1.07 | 0.12 |
| Model 26 | (100, 0.01, 0.9, 0.001) | 1.19 | 1.18 | 0.13 |
| Model 27 | (0, 0.2, 0.9, 0.001) | 1.00 | 0.99 | 0.13 |
| Model 28 | (0, 0.2, 0.9, 0.001) | 0.90 | 0.90 | 0.11 |
| Model 29 | (0, 0.05, 0.95, 0.001) | 0.97 | 1.00 | 0.11 |
| Model 30 | (100, 0.05, 0.95, 0.0003) | 1.01 | 0.98 | 0.15 |

Table 3: Evaluated models' configurations and validation performances

# E    Training, Validation, Testing set

Table 4 below shows the information of the instances contained in the training set, validation set and testing set for CSP. Column lists total number of instances in each dataset, while other columns list the number of instances with specific roll length n in that dataset. The division of VRPTW dataset can be found in Section 5.2.

Table 4: Dataset division for **CSP**

| Dataset | Total | $n = 50$ | $n = 100$ | $n = 200$ | $n = 750$ |
|---|---|---|---|---|---|
| Training | 400 | 160 | 160 | 80 | 0 |
| Validation | 30 | 10 | 10 | 10 | 0 |
| Testing | 156 | 49 | 0 | 86 | 21 |

# F    Curriculum Learning design

In Table 5, we provide data characteristic we considered, for the sake of curriculum learning of the RL agent. In this paper the RL agent is trained on instances that are ordered according to their difficulty level. To accomplish this instances are divided into three categories of easy, normal and hard according to the stock length for CSP.

Easy instances have stock length of 50, normal instances have stock length of 100 and hard instances have stock length of 200. There are 40 instances for each instance type. Details of training curriculum of different instance types are shown in table 5. Figure 8 displays the number of steps to convergence vs. instance number for training instances. It is clear that for each instance type the steps taken for convergence decreases as the model is trained on more of the same instance type. This shows that the RL agent successfully learns to select columns to enter basis. However, when instances are ordered randomly (Figure 8b) there are no specific trend on the steps taken to converge. This highlights the necessity of curriculum learning. Curriculum for VRPTW can be found in Section 5.2.

Table 5: Curriculum Learning schedule for **CSP**

| Training curriculum | | | |
|---|---|---|---|
| Type of Instance | Number of Instances | Stock Length | Number of Orders |
| Easy | 40 | 50 | 50 |
| Easy | 40 | 50 | 75 |
| Easy | 40 | 50 | 100 |
| Easy | 40 | 50 | 120 |
| Normal | 40 | 100 | 75 |
| Normal | 40 | 100 | 100 |
| Normal | 40 | 100 | 120 |
| Normal | 40 | 100 | 150 |
| Hard | 40 | 200 | 125 |
| Hard | 40 | 200 | 150 |

## G   Node Features

Node features used for **CSP**:

1. **Column node features**:
   Feature (a) and (c) relate to solving the RMP problem as they are all information about decision variables in RMP, and each column node corresponds to one decision variable. Feature (b) and (d) are determined by the problem formulation of each cutting stock instance, while feature (e) - (i) corresponds to the dynamical information of each column entering and leaving the basis.

   (a) **Reduced cost**: Reduced cost is a quantity associated with each variable indicating how much the objective function coefficient on the corresponding variable must be improved before the solution value of the decision variable will be positive in the optimal solution (the cutting pattern will be used in optimal set of cutting patterns). The reduced cost value is only non-zero when the optimal value of a variable is zero.

   (b) **Connectivity of column node**: Total number of constraint nodes each column node connects to. As each constraint is a particular demand, this node feature indicates the ability of a column node (a pattern) to satisfy demands. It also indicates the connectivity of each column node in the bipartite graph representing the state.

   (c) **Solution value**: The solution value of each decision variable corresponding to each column node after solving the RMP in the current iteration. For each column node, this feature is continuous number greater than or equal to 0. The candidate column nodes have this feature set to be 0.

   (d) **Waste**: A feature recording the remaining length of a roll if we were to cut the current pattern from the roll. Again, each column node corresponds to one decision variable in RMP, which also represents one particular cut pattern.

   (e) **Number of iterations that each column node stays in the basis**: If the column node stays in the basis for a long time, it is most likely that the pattern corresponds to this column node is really good.

(f) **Number of iterations that each column node stays out of the basis**: if the column node stays out of the basis for a long time, it is most likely never enters the basis and being used in optimal solution in future iterations.

(g) **If the column left basis on the last iteration or not**: This is a binary feature recording the dynamics of each column node.

(h) **If the column enter basis on the last iteration or not**: Similar binary feature as (f).

(i) **Action node or not**: A binary feature indicating whether a column node is a candidate (a newly added action) or not. If the column node is a candidate node (column) that is generated at the current iteration by SP, then this binary feature is 1 otherwise 0.

2. **Constraint node features**:
   Each constraint node corresponds to one constraint in RMP, so the number of constraint nodes are fixed for each cutting stock problem instance.

   (a) **Dual value**: Dual value or shadow price is the coefficient of each dual variable in sub-problem objective function, and as each constraint node corresponds to one dual variable, we record dual value as one feature for constraint node.

   (b) **Connectivity of constraint node**: Total number of column nodes each constraint node connects to, which also indicates the connectivity of each constraint node in the bipartite graph representing the state.

**VRPTW** node features used are quite similar to CSP:

1. **Column node features**: Reduced cost, connectivity of column node, solution value, route cost, Number of iterations that each column node stays in the basis, Number of iterations that each column node stays in the basis, If the column left basis on the last iteration or no, If the column enter basis on the last iteration or no.

2. **Constraint node features**: Dual value, connectivity of constraint node

## H  Detailed statistics of testing results

In Table 6, we provide statistics obtained from our experimental results for CSP. We report both the average and standard deviation of number of iterations, solution time measured in seconds, and the objective function values. Note that the resulting objective function values between column selection policies might differ due to the early-stopping criteria we adopted. Note that for the sake table spacing, we use Objval to denote objective function value, and $\mu, \sigma$ for mean and standard deviation, respectively. We observe the clear dominance of RLCG over the potential of RLCG in solving challenging CG problems in practice.

| | $n = 50$ | | | | | | $n = 200$ | | | | | | $n = 750$ | | | | | |
|---|---|---|---|---|---|---|---|---|---|---|---|---|---|---|---|---|---|---|
| | Iteration | | Time(s) | | Objval | | Iteration | | Time(s) | | Objval | | Iteration | | Time(s) | | Objval | |
| | $\mu$ | $\sigma$ | $\mu$ | $\sigma$ | $\mu$ | $\sigma$ | $\mu$ | $\sigma$ | $\mu$ | $\sigma$ | $\mu$ | $\sigma$ | $\mu$ | $\sigma$ | $\mu$ | $\sigma$ | $\mu$ | $\sigma$ |
| **Greedy** | 51.2 | 11.4 | 6.0 | 2.5 | 23.4 | 2.8 | 78.0 | 23.0 | 22.3 | 14.4 | 91.1 | 10.1 | 292.8 | 47.7 | 640.8 | 240.2 | 327.6 | 18.2 |
| **RL** | 41.6 | 10.0 | 4.9 | 2.2 | 23.5 | 2.8 | 64.7 | 20.5 | 18.1 | 12.3 | 91.5 | 9.8 | 227.2 | 42.7 | 460.3 | 219.9 | 328.6 | 18.2 |
| **Expert** | 42.5 | 9.5 | 7.1 | 3.1 | 23.4 | 2.8 | 69.1 | 18.8 | 29.3 | 18.9 | 91.2 | 10.1 | 251.3 | 35.2 | 827.9 | 302.0 | 327.2 | 18.7 |

Table 6: Solution time, iteration and objective function value reports with $\mu$ mean, and $\sigma$ standard deviation for CSP

In Table 7, we provide the same statistics obtained from our experimental results for VRPTW.

| | small test instances | | | | | | large test instances | | | | | |
|---|---|---|---|---|---|---|---|---|---|---|---|---|
| | Iteration | | Time(s) | | Objval | | Iteration | | Time(s) | | Objval | |
| | $\mu$ | $\sigma$ | $\mu$ | $\sigma$ | $\mu$ | $\sigma$ | $\mu$ | $\sigma$ | $\mu$ | $\sigma$ | $\mu$ | $\sigma$ |
| Greedy | 18.8 | 11.7 | 291.1 | 285.0 | 102.6 | 30 | 128.1 | 100.3 | 3268.2 | 4627.5 | 458.6 | 144.6 |
| RL | 9.5 | 6.3 | 179.9 | 247.1 | 102.6 | 30.0 | 75.6 | 39.3 | 1832.8 | 2015.3 | 458.6 | 144.6 |

Table 7: Solution time, iteration and objective function value reports with $\mu$ mean, and $\sigma$ standard deviation for **VRPTW**

# I   Per instance results CSP

| Names | Iterations | | Time | | Objective value | |
|---|---|---|---|---|---|---|
| Instance name | Greedy_iter | RL_iter | Greedy_time | RL_time | Greedy_obj | RL_obj |
| r107.txt, n = 20 | 328 | 84 | 15164.16 | 3843.84 | 517.00 | 517.00 |
| r110.txt, n = 19 | 189 | 101 | 4093.69 | 2170.76 | 522.00 | 522.00 |
| c101.txt, n = 25 | 140 | 90 | 4123.05 | 2632.07 | 637.00 | 637.00 |
| r106.txt, n = 20 | 56 | 30 | 807.00 | 429.26 | 637.00 | 637.00 |
| r111.txt, n = 20 | 203 | 126 | 6965.91 | 4350.76 | 534.00 | 534.00 |
| r104.txt, n = 20 | 261 | 119 | 18788.82 | 8359.52 | 504.00 | 504.00 |
| r108.txt, n = 18 | 49 | 39 | 6062.88 | 4529.36 | 156.00 | 156.00 |
| r103.txt, n = 19 | 440 | 142 | 15789.96 | 5132.63 | 437.00 | 437.00 |
| rc104.txt, n = 15 | 196 | 79 | 2992.44 | 1211.46 | 491.00 | 491.00 |
| rc103.txt, n = 19 | 163 | 84 | 2135.99 | 1106.87 | 667.00 | 667.00 |
| rc107.txt, n = 15 | 77 | 77 | 766.64 | 783.29 | 341.00 | 341.00 |
| rc105.txt, n = 15 | 107 | 88 | 578.83 | 497.95 | 415.00 | 415.00 |
| rc101.txt, n = 16 | 40 | 27 | 106.21 | 69.52 | 571.00 | 571.00 |
| rc102.txt, n = 14 | 115 | 73 | 999.08 | 645.00 | 391.00 | 391.00 |
| rc108.txt, n = 15 | 345 | 173 | 5449.42 | 2649.93 | 383.00 | 383.00 |
| r102.txt, n = 14 | 110 | 59 | 960.38 | 513.88 | 382.00 | 382.00 |
| c101.txt, n = 15 | 36 | 32 | 88.32 | 78.41 | 491.00 | 491.00 |
| r105.txt, n = 15 | 173 | 93 | 3877.94 | 2100.79 | 583.00 | 583.00 |
| c107.txt, n = 15 | 93 | 83 | 5609.29 | 4938.09 | 249.00 | 249.00 |
| c105.txt, n = 20 | 87 | 81 | 3874.88 | 3609.85 | 485.00 | 485.00 |
| rc201.txt, n = 16 | 57 | 51 | 1752.45 | 1567.67 | 641.00 | 641.00 |
| rc106.txt, n = 15 | 128 | 98 | 991.33 | 745.47 | 276.00 | 276.00 |
| r101.txt, n = 15 | 63 | 55 | 100.00 | 87.98 | 409.00 | 409.00 |
| r205.txt, n = 19 | 11 | 16 | 1498.44 | 2213.35 | 749.00 | 749.00 |
| r108.txt, n = 15 | 37 | 30 | 2992.61 | 2321.59 | 148.75 | 148.75 |
| r109.txt, n = 19 | 153 | 71 | 1294.90 | 603.83 | 578.00 | 578.00 |

# J   VRPTW box plots

Similar to CSP, we see the upward generalization of `RLCG` in VRPTW: we only train our model with small sized instances, and tested on large instances with more customers, and we observe that `RLCG` was able to perform well using VRPTW testing instances with large size. These results, again indicate that our proposed `RLCG` can be preferable in solving challenging column generation problems because of its ability to generalize well. Besides, compare large testing results with small testing results for VRPTW (also compare n=750 results with n=50,200 for CSP), we observe that the more challenging the problem, the larger gap exist between `RLCG` and benchmark method. This indicates that for CG problem we considered, the harder the problem, the more important considering the future effect of CG becomes, thus taking future into consideration would drastically accelerate the solving process of CG.

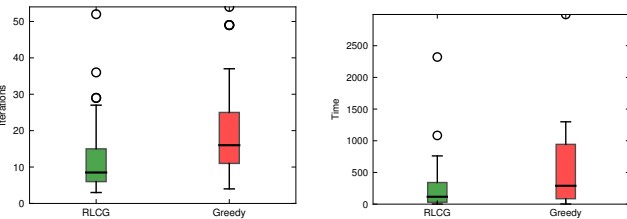

Figure 11: Pair-wise comparisons between RLCG and greedy benchmark in terms of the number of CG iterations and the solving time over small and large testing sets for **VRPTW**

| Name | Iterations | | | Time | | | Objective value | | |
|---|---|---|---|---|---|---|---|---|---|
| Instance name | Greedy_iter | RL_iter | Expert_iter | Greedy_time | RL_time | Expert_time | Greedy_obj | RL_obj | Expert_obj |
| BPP_50_125_0.1_0.7_2 | 45 | 30 | 28 | 4.45 | 2.80 | 3.36 | 18.18 | 18.32 | 18.31 |
| BPP_50_200_0.2_0.8_5 | 46 | 31 | 31 | 5.15 | 3.46 | 5.12 | 29.00 | 29.00 | 29.00 |
| BPP_50_50_0.1_0.8_0 | 40 | 31 | 31 | 2.64 | 2.22 | 2.81 | 23.50 | 23.50 | 23.50 |
| BPP_50_125_0.1_0.8_8 | 53 | 38 | 48 | 6.04 | 4.31 | 7.84 | 20.74 | 21.31 | 20.64 |
| BPP_50_200_0.2_0.7_2 | 57 | 42 | 38 | 6.70 | 5.10 | 6.47 | 23.06 | 23.00 | 23.00 |
| BPP_50_125_0.1_0.7_8 | 57 | 52 | 47 | 6.13 | 5.65 | 6.82 | 20.04 | 20.31 | 20.18 |
| BPP_50_200_0.2_0.7_0 | 43 | 42 | 35 | 4.48 | 4.85 | 5.50 | 23.46 | 23.58 | 23.45 |
| BPP_50_125_0.1_0.8_6 | 61 | 43 | 57 | 8.35 | 5.71 | 11.61 | 25.25 | 25.50 | 25.25 |
| BPP_50_200_0.2_0.8_7 | 57 | 44 | 50 | 6.08 | 4.83 | 7.93 | 24.00 | 23.96 | 23.94 |
| BPP_50_125_0.1_0.8_1 | 67 | 54 | 49 | 7.64 | 6.36 | 7.83 | 22.67 | 22.79 | 22.68 |
| BPP_50_50_0.1_0.8_1 | 25 | 23 | 23 | 1.50 | 1.56 | 1.92 | 28.00 | 28.00 | 28.00 |
| BPP_50_200_0.2_0.7_3 | 63 | 45 | 54 | 7.91 | 5.44 | 9.92 | 23.50 | 23.50 | 23.50 |
| BPP_50_125_0.2_0.7_4 | 44 | 37 | 31 | 3.08 | 2.80 | 3.02 | 25.00 | 25.00 | 25.00 |
| BPP_50_200_0.2_0.7_5 | 68 | 52 | 46 | 9.80 | 7.33 | 9.08 | 24.60 | 24.70 | 24.70 |
| BPP_50_125_0.1_0.8_3 | 35 | 29 | 34 | 3.42 | 2.90 | 4.97 | 28.50 | 28.50 | 28.50 |
| BPP_50_125_0.1_0.8_2 | 59 | 55 | 52 | 9.66 | 9.25 | 12.15 | 20.29 | 20.51 | 20.33 |
| BPP_50_125_0.1_0.7_1 | 46 | 48 | 37 | 5.11 | 5.79 | 5.79 | 18.92 | 18.95 | 18.98 |
| BPP_50_200_0.2_0.7_8 | 56 | 31 | 34 | 7.96 | 3.94 | 6.69 | 21.65 | 21.83 | 21.62 |
| BPP_50_200_0.2_0.7_9 | 50 | 38 | 42 | 5.37 | 4.00 | 6.39 | 23.21 | 23.56 | 23.15 |
| BPP_50_200_0.1_0.8_1 | 62 | 56 | 59 | 8.62 | 8.05 | 12.62 | 23.21 | 23.25 | 23.07 |
| BPP_50_200_0.1_0.8_4 | 58 | 52 | 46 | 8.05 | 7.25 | 9.00 | 25.00 | 25.00 | 25.00 |
| BPP_50_125_0.1_0.7_4 | 51 | 42 | 41 | 6.13 | 5.01 | 6.78 | 21.30 | 21.33 | 21.42 |
| BPP_50_125_0.2_0.7_0 | 44 | 41 | 46 | 3.90 | 3.91 | 6.26 | 21.38 | 21.62 | 21.38 |
| BPP_50_200_0.1_0.8_6 | 62 | 50 | 53 | 8.99 | 6.96 | 10.77 | 22.60 | 22.86 | 22.58 |
| BPP_50_200_0.1_0.8_0 | 84 | 69 | 65 | 14.12 | 11.63 | 14.99 | 23.17 | 23.57 | 23.15 |
| BPP_50_125_0.1_0.7_5 | 45 | 28 | 36 | 4.90 | 2.87 | 5.43 | 19.71 | 20.04 | 19.88 |
| BPP_50_200_0.2_0.8_2 | 53 | 48 | 47 | 6.58 | 6.21 | 8.93 | 26.00 | 26.00 | 26.00 |
| BPP_50_50_0.1_0.8_2 | 31 | 29 | 30 | 1.62 | 1.66 | 2.06 | 25.00 | 25.00 | 25.00 |
| BPP_50_125_0.2_0.7_2 | 46 | 38 | 39 | 3.98 | 3.29 | 4.66 | 21.74 | 22.07 | 21.89 |
| BPP_50_50_0.1_0.8_4 | 39 | 35 | 34 | 2.56 | 2.53 | 3.21 | 23.25 | 23.29 | 23.25 |
| BPP_50_200_0.2_0.7_4 | 55 | 33 | 39 | 5.51 | 3.17 | 5.24 | 21.83 | 21.92 | 21.83 |
| BPP_50_125_0.1_0.8_7 | 50 | 43 | 46 | 5.70 | 5.20 | 8.07 | 26.00 | 26.00 | 26.00 |
| BPP_50_200_0.1_0.8_7 | 59 | 75 | 66 | 8.85 | 12.93 | 15.27 | 21.36 | 21.21 | 21.19 |
| BPP_50_125_0.2_0.7_3 | 46 | 39 | 38 | 3.52 | 3.19 | 3.95 | 23.00 | 23.00 | 23.00 |
| BPP_50_125_0.1_0.8_2 | 39 | 40 | 42 | 4.14 | 4.73 | 7.02 | 20.80 | 20.82 | 20.62 |
| BPP_50_125_0.1_0.7_7 | 47 | 42 | 41 | 5.65 | 5.04 | 6.99 | 19.63 | 19.81 | 19.70 |
| BPP_50_125_0.1_0.8_5 | 40 | 36 | 35 | 3.67 | 3.46 | 4.66 | 29.50 | 29.50 | 29.50 |
| BPP_50_125_0.2_0.7_1 | 61 | 44 | 48 | 7.03 | 4.97 | 7.88 | 23.15 | 23.15 | 23.15 |
| BPP_50_200_0.2_0.8_0 | 70 | 42 | 51 | 10.33 | 5.75 | 10.72 | 27.50 | 27.50 | 27.50 |
| BPP_50_125_0.1_0.8_9 | 50 | 43 | 48 | 5.94 | 5.35 | 8.84 | 23.50 | 23.50 | 23.50 |
| BPP_50_125_0.1_0.7_6 | 52 | 40 | 40 | 6.01 | 4.64 | 6.51 | 20.77 | 20.98 | 20.90 |
| BPP_50_200_0.1_0.8_9 | 64 | 42 | 52 | 8.31 | 5.28 | 9.66 | 21.88 | 21.99 | 21.83 |
| BPP_50_50_0.1_0.8_3 | 30 | 30 | 26 | 1.65 | 1.76 | 1.75 | 20.06 | 20.12 | 20.07 |
| BPP_50_125_0.1_0.7_9 | 51 | 42 | 43 | 6.26 | 5.10 | 7.23 | 20.46 | 20.66 | 20.59 |
| BPP_50_125_0.1_0.8_0 | 49 | 41 | 46 | 5.32 | 4.70 | 7.57 | 26.00 | 26.00 | 26.00 |
| BPP_50_200_0.1_0.8_3 | 66 | 45 | 42 | 8.79 | 5.84 | 7.68 | 21.98 | 22.14 | 22.21 |
| BPP_50_200_0.1_0.8_5 | 34 | 33 | 35 | 3.48 | 3.57 | 5.60 | 30.00 | 30.00 | 30.00 |
| BPP_50_200_0.2_0.7_7 | 57 | 48 | 50 | 6.51 | 5.61 | 8.30 | 23.50 | 23.50 | 23.50 |
| BPP_50_200_0.2_0.8_1 | 41 | 28 | 33 | 4.82 | 3.32 | 6.06 | 27.50 | 27.50 | 27.50 |

# K   per instance results CSP

| Name | Iterations | | | Time | | | Objective value | | |
|---|---|---|---|---|---|---|---|---|---|
| Instance name | Greedy_iter | RL_iter | Expert_iter | Greedy_time | RL_time | Expert_time | Greedy_obj | RL_obj | Expert_obj |
| BPP_200_100_0.2_0.7_1 | 57 | 45 | 56 | 11.86 | 9.18 | 18.19 | 90.50 | 90.81 | 90.50 |
| BPP_200_100_0.1_0.7_4 | 80 | 61 | 68 | 24.36 | 18.23 | 31.03 | 79.53 | 79.93 | 79.70 |
| BPP_200_75_0.2_0.8_5 | 63 | 52 | 52 | 10.91 | 9.10 | 13.37 | 103.00 | 103.00 | 103.00 |
| BPP_200_75_0.2_0.7_9 | 55 | 47 | 52 | 7.55 | 6.43 | 10.35 | 89.24 | 89.67 | 89.25 |
| BPP_200_75_0.1_0.7_4 | 66 | 48 | 67 | 12.67 | 8.93 | 19.91 | 80.44 | 81.72 | 80.37 |
| BPP_200_100_0.2_0.7_0 | 65 | 53 | 59 | 13.45 | 11.01 | 19.05 | 93.45 | 93.40 | 93.40 |
| BPP_200_100_0.1_0.7_5 | 69 | 49 | 71 | 19.29 | 13.06 | 31.68 | 78.27 | 79.52 | 78.02 |
| BPP_200_50_0.2_0.8_1 | 30 | 30 | 29 | 2.42 | 2.70 | 3.44 | 115.50 | 115.50 | 115.50 |
| BPP_200_120_0.1_0.7_7 | 87 | 72 | 87 | 33.18 | 26.98 | 52.32 | 78.47 | 79.01 | 78.47 |
| BPP_200_50_0.2_0.8_5 | 44 | 32 | 42 | 3.92 | 2.93 | 5.46 | 103.00 | 103.00 | 103.00 |
| BPP_200_100_0.1_0.7_7 | 71 | 56 | 65 | 20.23 | 15.24 | 28.03 | 77.10 | 77.60 | 77.03 |
| BPP_200_75_0.1_0.7_7 | 78 | 57 | 64 | 16.39 | 11.12 | 18.52 | 82.80 | 83.70 | 82.92 |
| BPP_200_75_0.1_0.7_1 | 75 | 63 | 69 | 15.37 | 12.84 | 21.38 | 82.69 | 83.60 | 82.64 |
| BPP_200_75_0.1_0.8_7 | 97 | 82 | 82 | 26.72 | 21.65 | 31.89 | 91.71 | 92.08 | 91.72 |
| BPP_200_75_0.2_0.8_0 | 59 | 48 | 59 | 9.94 | 8.15 | 15.58 | 102.33 | 102.33 | 102.33 |
| BPP_200_120_0.1_0.7_3 | 111 | 89 | 92 | 51.56 | 39.32 | 63.39 | 82.81 | 83.30 | 83.24 |
| BPP_200_100_0.2_0.8_8 | 85 | 67 | 73 | 24.35 | 18.42 | 31.85 | 111.00 | 111.00 | 111.00 |
| BPP_200_100_0.2_0.7_8 | 88 | 51 | 75 | 20.32 | 10.12 | 25.20 | 91.21 | 92.33 | 91.19 |
| BPP_200_100_0.2_0.8_2 | 107 | 89 | 88 | 33.04 | 26.74 | 39.09 | 99.56 | 99.59 | 99.56 |
| BPP_200_75_0.2_0.7_2 | 56 | 46 | 49 | 7.75 | 6.12 | 9.31 | 92.30 | 92.96 | 92.44 |
| BPP_200_75_0.1_0.8_9 | 84 | 87 | 87 | 20.47 | 23.16 | 33.23 | 92.00 | 92.28 | 92.00 |
| BPP_200_100_0.1_0.8_4 | 106 | 87 | 85 | 43.36 | 34.82 | 51.75 | 94.50 | 94.50 | 94.50 |
| BPP_200_100_0.1_0.8_6 | 105 | 84 | 87 | 39.42 | 30.47 | 47.84 | 88.96 | 89.42 | 89.18 |
| BPP_200_75_0.1_0.8_4 | 84 | 82 | 75 | 21.36 | 21.97 | 28.12 | 92.03 | 92.06 | 92.01 |
| BPP_200_100_0.2_0.8_9 | 121 | 92 | 89 | 40.25 | 28.35 | 40.78 | 101.94 | 102.00 | 101.87 |
| BPP_200_75_0.2_0.8_8 | 75 | 55 | 59 | 13.82 | 9.63 | 15.57 | 101.50 | 101.50 | 101.50 |
| BPP_200_100_0.1_0.8_8 | 124 | 112 | 106 | 56.95 | 52.58 | 73.34 | 93.24 | 93.29 | 93.58 |
| BPP_200_75_0.1_0.7_2 | 60 | 48 | 54 | 11.35 | 8.97 | 15.03 | 78.03 | 78.43 | 78.03 |
| BPP_200_75_0.2_0.8_2 | 71 | 55 | 61 | 12.99 | 9.86 | 16.48 | 96.65 | 96.58 | 96.58 |
| BPP_200_75_0.2_0.7_7 | 54 | 50 | 50 | 7.37 | 6.85 | 9.84 | 91.00 | 91.00 | 91.00 |
| BPP_200_100_0.1_0.7_9 | 77 | 70 | 77 | 24.27 | 22.05 | 37.97 | 80.15 | 80.62 | 80.34 |
| BPP_200_75_0.1_0.7_9 | 58 | 54 | 65 | 10.46 | 9.92 | 18.38 | 82.59 | 83.25 | 82.41 |
| BPP_200_100_0.2_0.8_6 | 110 | 93 | 99 | 36.26 | 29.91 | 48.23 | 98.00 | 98.00 | 98.00 |
| BPP_200_75_0.2_0.8_3 | 61 | 50 | 58 | 10.39 | 8.45 | 15.06 | 102.50 | 102.50 | 102.50 |
| BPP_200_100_0.1_0.7_2 | 95 | 69 | 72 | 31.75 | 20.94 | 33.39 | 80.63 | 81.30 | 81.02 |
| BPP_200_75_0.2_0.7_8 | 56 | 39 | 50 | 7.73 | 4.96 | 9.78 | 89.12 | 89.70 | 89.12 |
| BPP_200_100_0.1_0.7_8 | 85 | 74 | 85 | 27.10 | 23.43 | 41.74 | 80.91 | 81.01 | 80.70 |
| BPP_200_100_0.1_0.8_5 | 102 | 86 | 95 | 42.48 | 34.58 | 61.02 | 90.08 | 91.64 | 89.99 |
| BPP_200_100_0.1_0.7_3 | 80 | 67 | 65 | 22.67 | 18.24 | 26.34 | 79.88 | 80.18 | 79.66 |
| BPP_200_75_0.2_0.8_6 | 54 | 41 | 43 | 8.81 | 6.63 | 10.44 | 115.00 | 115.00 | 115.00 |
| BPP_200_75_0.2_0.8_7 | 57 | 52 | 54 | 9.08 | 8.60 | 13.34 | 108.00 | 108.00 | 108.00 |
| BPP_200_120_0.1_0.7_0 | 104 | 84 | 81 | 44.38 | 34.28 | 50.55 | 80.17 | 80.86 | 80.89 |
| BPP_200_75_0.1_0.7_4 | 66 | 48 | 67 | 12.58 | 8.78 | 19.99 | 80.44 | 81.72 | 80.37 |
| BPP_200_100_0.2_0.7_7 | 87 | 72 | 78 | 21.06 | 16.82 | 27.69 | 93.57 | 93.69 | 93.50 |
| BPP_200_120_0.1_0.7_8 | 69 | 53 | 73 | 22.31 | 16.40 | 38.58 | 79.73 | 80.62 | 79.57 |
| BPP_200_100_0.1_0.8_3 | 106 | 93 | 88 | 40.46 | 35.01 | 49.13 | 89.03 | 89.98 | 89.15 |
| BPP_200_75_0.1_0.8_8 | 87 | 81 | 81 | 23.46 | 22.07 | 32.42 | 87.02 | 87.65 | 87.03 |
| BPP_200_100_0.1_0.7_0 | 80 | 72 | 74 | 23.89 | 21.35 | 33.65 | 81.23 | 81.34 | 80.98 |
| BPP_200_50_0.2_0.8_2 | 35 | 29 | 36 | 3.00 | 3.35 | 4.57 | 103.50 | 103.50 | 103.50 |
| BPP_200_100_0.1_0.8_1 | 99 | 102 | 102 | 48.69 | 52.01 | 81.43 | 84.03 | 84.11 | 83.68 |
| BPP_200_120_0.1_0.7_6 | 88 | 75 | 80 | 31.71 | 26.87 | 44.49 | 82.98 | 83.25 | 83.06 |
| BPP_200_120_0.1_0.7_5 | 95 | 78 | 78 | 37.97 | 30.99 | 46.90 | 80.48 | 80.68 | 80.82 |
| BPP_200_100_0.2_0.8_3 | 97 | 77 | 84 | 29.38 | 22.22 | 38.00 | 100.33 | 100.33 | 100.33 |
| BPP_200_50_0.2_0.8_3 | 40 | 33 | 34 | 3.54 | 3.04 | 4.16 | 101.83 | 101.83 | 101.83 |
| BPP_200_100_0.2_0.8_1 | 87 | 59 | 73 | 25.16 | 15.68 | 31.66 | 106.00 | 106.00 | 106.00 |
| BPP_200_120_0.1_0.7_9 | 94 | 80 | 76 | 36.10 | 30.36 | 51.57 | 79.99 | 80.27 | 79.99 |

| Name | Iterations | | | Time | | | Objective value | | |
|---|---|---|---|---|---|---|---|---|---|
| Instance name | Greedy_iter | RL_iter | Expert_iter | Greedy_time | RL_time | Expert_time | Greedy_obj | RL_obj | Expert_obj |
| BPP_200_75_0.2_0.7_6 | 52 | 45 | 44 | 6.17 | 5.82 | 7.85 | 94.45 | 94.45 | 94.45 |
| BPP_200_75_0.1_0.8_5 | 99 | 84 | 74 | 27.22 | 22.59 | 28.18 | 89.04 | 89.48 | 89.34 |
| BPP_200_50_0.2_0.8_0 | 41 | 32 | 37 | 3.57 | 2.90 | 4.70 | 98.33 | 98.33 | 98.33 |
| BPP_200_50_0.2_0.8_9 | 36 | 28 | 33 | 3.06 | 2.44 | 4.16 | 108.50 | 108.50 | 108.50 |
| BPP_200_120_0.1_0.7_2 | 107 | 82 | 91 | 46.05 | 32.88 | 58.13 | 80.49 | 81.01 | 80.66 |
| BPP_200_50_0.2_0.8_6 | 37 | 30 | 36 | 3.23 | 2.64 | 4.54 | 101.50 | 101.50 | 101.50 |
| BPP_200_50_0.2_0.8_4 | 38 | 31 | 34 | 3.30 | 2.88 | 4.24 | 105.50 | 105.50 | 105.50 |
| BPP_200_50_0.2_0.8_7 | 36 | 35 | 32 | 3.00 | 3.23 | 3.90 | 102.00 | 102.00 | 102.00 |
| BPP_200_75_0.1_0.8_2 | 96 | 80 | 82 | 25.07 | 20.13 | 30.15 | 87.01 | 87.31 | 87.02 |
| BPP_200_100_0.1_0.7_6 | 97 | 72 | 81 | 31.30 | 21.39 | 37.84 | 81.77 | 82.77 | 82.23 |
| BPP_200_100_0.1_0.8_0 | 126 | 111 | 109 | 56.03 | 48.24 | 70.80 | 92.00 | 92.67 | 92.17 |
| BPP_200_75_0.1_0.7_0 | 73 | 59 | 68 | 14.63 | 11.65 | 20.31 | 83.04 | 83.35 | 82.97 |
| BPP_200_100_0.2_0.8_4 | 78 | 66 | 70 | 21.59 | 18.17 | 30.14 | 112.00 | 112.00 | 112.00 |
| BPP_200_75_0.1_0.8_4 | 87 | 65 | 75 | 17.02 | 12.03 | 20.98 | 102.73 | 102.73 | 102.73 |
| BPP_200_75_0.2_0.7_0 | 70 | 59 | 61 | 10.48 | 8.86 | 12.56 | 89.60 | 90.05 | 89.65 |
| BPP_200_75_0.1_0.8_6 | 81 | 72 | 74 | 20.60 | 17.78 | 27.62 | 84.92 | 85.28 | 84.83 |
| BPP_200_75_0.1_0.8_0 | 89 | 80 | 83 | 24.40 | 21.46 | 32.58 | 88.28 | 88.92 | 88.43 |
| BPP_200_75_0.1_0.7_8 | 56 | 52 | 64 | 10.33 | 9.84 | 19.50 | 77.35 | 77.48 | 77.21 |
| BPP_200_75_0.2_0.8_1 | 87 | 74 | 72 | 17.77 | 14.93 | 20.73 | 101.83 | 101.83 | 101.92 |
| BPP_200_75_0.1_0.7_6 | 74 | 61 | 57 | 14.43 | 12.27 | 16.06 | 82.75 | 82.68 | 82.81 |
| BPP_200_75_0.1_0.8_3 | 59 | 59 | 57 | 13.11 | 13.84 | 19.95 | 105.50 | 105.50 | 105.50 |
| BPP_200_120_0.1_0.7_1 | 104 | 86 | 84 | 43.51 | 34.34 | 51.44 | 82.28 | 82.78 | 82.28 |
| BPP_200_75_0.2_0.8_9 | 67 | 64 | 63 | 21.74 | 12.43 | 17.49 | 94.89 | 94.92 | 94.89 |
| BPP_200_120_0.1_0.8_3 | 118 | 100 | 104 | 61.68 | 50.27 | 81.97 | 89.24 | 89.80 | 89.14 |
| BPP_200_120_0.1_0.7_4 | 96 | 60 | 73 | 41.30 | 22.68 | 45.75 | 78.44 | 79.64 | 78.58 |
| BPP_200_100_0.2_0.7_9 | 97 | 86 | 79 | 23.71 | 21.21 | 27.41 | 94.59 | 94.92 | 94.59 |
| BPP_200_100_0.1_0.8_1 | 117 | 112 | 112 | 48.07 | 46.64 | 69.57 | 92.69 | 92.90 | 92.97 |
| BPP_200_100_0.1_0.7_1 | 69 | 65 | 61 | 20.25 | 19.44 | 27.35 | 78.83 | 78.92 | 79.06 |
| BPP_200_75_0.1_0.7_4 | 66 | 48 | 67 | 12.80 | 8.78 | 19.97 | 80.44 | 81.72 | 80.37 |
| BPP_200_75_0.2_0.7_1 | 59 | 43 | 46 | 8.50 | 5.70 | 9.00 | 86.87 | 87.82 | 86.80 |
| BPP_750_300_0.1_0.7_7 | 230 | 215 | 231 | 465.58 | 431.99 | 761.27 | 299.97 | 299.85 | 298.28 |
| BPP_750_300_0.1_0.7_8 | 288 | 215 | 252 | 642.50 | 438.36 | 868.59 | 301.65 | 302.74 | 301.51 |
| BPP_750_300_0.1_0.8_2 | 385 | 329 | 331 | 1222.12 | 984.78 | 1575.72 | 344.83 | 346.43 | 344.20 |
| BPP_750_300_0.1_0.8_3 | 330 | 327 | 316 | 987.16 | 992.71 | 1487.61 | 333.55 | 334.33 | 333.36 |
| BPP_750_300_0.1_0.8_7 | 397 | 317 | 321 | 1238.79 | 900.67 | 1466.03 | 334.83 | 336.11 | 335.26 |
| BPP_750_300_0.2_0.7_6 | 377 | 194 | 226 | 924.06 | 348.74 | 695.31 | 339.59 | 341.83 | 343.53 |
| BPP_750_300_0.2_0.7_7 | 302 | 202 | 241 | 652.23 | 374.72 | 761.56 | 346.00 | 346.91 | 345.16 |
| BPP_750_300_0.2_0.7_8 | 266 | 190 | 213 | 555.75 | 343.14 | 643.76 | 339.14 | 340.80 | 340.28 |
| BPP_750_300_0.1_0.7_5 | 262 | 225 | 243 | 676.38 | 553.77 | 991.46 | 302.62 | 305.25 | 302.32 |
| BPP_750_300_0.1_0.7_2 | 282 | 218 | 225 | 607.61 | 424.71 | 723.86 | 308.22 | 310.59 | 307.71 |
| BPP_750_300_0.2_0.7_1 | 251 | 215 | 213 | 393.61 | 324.76 | 513.24 | 343.90 | 343.73 | 344.31 |
| BPP_750_300_0.1_0.7_4 | 252 | 213 | 248 | 526.86 | 424.10 | 839.43 | 299.13 | 301.11 | 298.10 |
| BPP_750_300_0.1_0.7_9 | 251 | 266 | 255 | 518.30 | 576.85 | 867.30 | 303.05 | 301.99 | 301.48 |
| BPP_750_300_0.2_0.7_0 | 264 | 199 | 240 | 436.00 | 291.99 | 602.91 | 338.58 | 337.66 | 336.33 |
| BPP_750_300_0.2_0.7_2 | 330 | 206 | 284 | 595.43 | 289.76 | 719.85 | 344.99 | 347.38 | 345.10 |
| BPP_750_300_0.2_0.7_3 | 291 | 223 | 258 | 496.82 | 347.74 | 667.53 | 333.32 | 332.97 | 333.40 |
| BPP_750_300_0.2_0.7_4 | 304 | 201 | 224 | 539.88 | 299.66 | 567.41 | 343.59 | 343.89 | 343.54 |
| BPP_750_300_0.1_0.7_6 | 254 | 221 | 224 | 545.02 | 452.20 | 755.89 | 302.22 | 303.04 | 300.90 |
| BPP_750_300_0.2_0.7_0 | 264 | 199 | 240 | 437.81 | 294.21 | 618.61 | 338.58 | 337.66 | 336.33 |
| BPP_750_300_0.2_0.7_2 | 330 | 206 | 284 | 595.81 | 298.91 | 749.31 | 344.99 | 347.38 | 345.10 |
| BPP_750_300_0.2_0.7_5 | 239 | 190 | 209 | 398.43 | 271.92 | 508.35 | 335.84 | 337.90 | 335.33 |