# OpenReview forum: "A Deep Reinforcement Learning Framework for Column Generation"
_NeurIPS.cc/2022/Conference — NeurIPS 2022 Accept_

### Official Review · Reviewer_Sm39 · 2022-06-23

**Rating:** 6
**Confidence:** 3
**Soundness:** 3 good
**Presentation:** 2 fair
**Contribution:** 3 good

**Summary:**

This paper studies the use of reinforcement learning to improve the column generation algorithm. In particular, the authors design a state space around a bipartite graph containing column nodes and constraint nodes, and an action space derived from RMP and SP formulations of the problem. These two aspects are the main challenge of the translation, the reward function is then set to a reasonable choice, and a standard RL algorithm (DQN) is applied. Testing in two problems (cutting stock and vehicle routing) show their method outperforms a baseline.

**Questions:**

Questions:
* Sec. 3: I have trouble understanding how the SP formulation follows from the RMP formulation (Line 130). The transition here is very abrupt, where does \pi come from, how is it computed from the RMP, etc. A figure (conceptual example) would help here a lot as well.
* L165-168: You design column node features and constraint node features based “based on our previous experience”. Are these features problem independent? How vital are they for your method? Can you really defer this to the appendix?
* L 172: It is unclear from the text how the exact GNN looks like. I do understand the bipartite graph structure, but how are all combined node embeddings mapped to the Q-function output? You need to at least briefly mention something about this in the main paper text.



Conclusion: This is a decent paper. The main contribution of the paper is the adaptation of CG to the RL setting. The main challenge is the way to represent the state space (as a bipartite graph), and the use of RMP and SP formulations to design an action space at each iteration. They can then apply any standard RL method, like DQN in this paper. The method subsequently outperforms the baseline. There have been many new applications of RL to combinatorial optimization problems in recent years (which makes the paper less groundbreaking), but it does seem to be a valuable addition.

**Limitations:**

-

**Strengths And Weaknesses:**

Strong:
* The application of RL to combinatorial optimization is a relevant research topic that gets more and more attention.
* The method outperforms the baseline on two relevant problems (cutting stock problem and vehicle routing problem with time windows).
* I like Figure 1, it clarifies a lot.
* You open source your code.

Weak:
* See questions section.
* There are very little details on the RL side of things (like a DQN update equation or target computation). I understand your focus is on the adaptation of column generation to RL, but I do think one or two paragraph with an equation would help a lot.
* You repeatedly point to Figure 6 in Appendix A, but if that is important to explain your method, it should appear in the main paper.
In general you have quite a lot of text and little conceptual illustration. I think a good Figure 1 showing your method could help a lot.

---

> ### Author Response · Authors · 2022-08-02
> **Comments to Reviewer Sm39**
>
> Thank you very much for your careful review and constructive comments, which have much contributed to improving the paper. We will address all the comments in the paper and provide responses to the comments below.
>
> ### SP and RMP formulations:
> The master problem (RMP) is the main problem we are trying to optimize, and the subproblem (SP) is used to generate good quality patterns that can be added to the RMP. In the subproblem, instead of explicitly searching over all patterns (represented by decision variables x) and returning the most negative reduced cost pattern, we search for the most negative reduced cost pattern by the optimization shown in the subproblem formulation in the paper (we made a typo, should be max instead of min for the subproblem formulation). \pi are the shadow prices associated with the RMP constraints, which we get from solving the current RMP problem. Once the current RMP is solved, we can use the SP formulation to generate patterns. The objective of the subproblem is directly minimizing the reduced cost, so it will return the pattern or the column with the most negative reduced cost, which we add to the next iteration RMP in the conventional CG process.
>
> ### Column node features and constraint node features:
> The features we selected can be grouped into two types:
> 1) Static features are about solving RMP, SP at each iteration:  reduced costs, solution values, dual values, and connectivity of nodes
> 2) Dynamic features related to the overall CG solving history:  Number of iterations that each column node stays in the basis, Number of iterations that each column node stays out of basis, If the column left basis on the last iteration or no, If the column enter basis on the last iteration or not
> Our features are problem independent and therefore our RLCG can be applied to any CG problem. We have not extensively explored the feature space but these seem to be the most reasonable features. We will revise the main section to include more information about the features.
>
> ### GNN structure:
> We will add the following clarification to the appendix section E and a brief summary in the main text: We opt for a simple GNN based on the MPNN message-passing architecture [Gimler 2017]. MPNN has been used successfully in related work on ML for optimization, e.g., [Morabit 2021, Gasse 2019].
>
> In the MPNN, each node Vt has its own hidden state ht. For each node at each iteration out of K iterations, we have two operations:
>
> Message passing: we aggregate hidden states of all neighbouring nodes of the node Vt to obtain a message $m_{t+1}$ passed to node Vt.
>     $m_v^{t+1} = \sum_{w \in N(v)} M(h_v^t, h_w^t)$
>
> Hidden state update:  Then, we update the hidden state of the node Vt using the obtained message and the previous hidden state of that node.
>
> $h_v^{t+1} =  U(h_v^t, m_v^{t+1})$
>
> After K iterations, the final output for each node is given by:
>
> $\hat{y} = R(\{h_v^T | v \in G\})$
>
> where functions R, U, and M are all learned through the DQN training process. The output values are the Q values for that node, which is the total future expected reward if we add or keep that particular node in the next RMP iteration. For instance, node v6 in figure 1 would have higher Q values than node v7. N(v) represents the neighbours of node V following the bipartite structure of our RMP.
>
> In our implementation, M is composed of 2 dense layers, U is composed of 2 dense layers, and R is composed of 3 dense layers. All layers use ReLu activation functions. We use K = 4 in our GNN.

---

### Official Review · Reviewer_8Csh · 2022-07-08

**Rating:** 5
**Confidence:** 4
**Soundness:** 3 good
**Presentation:** 2 fair
**Contribution:** 3 good

**Summary:**

This work focuses on using Column Generation (CG) as an algorithm to solve large-scale linear programs. A DQN algorithm (called RLCG) combined with Graph Neural Networks is used to improve the column generation procedure. Specifically, RL is used to select columns for linear programs with many variables. To do so, it assumes that the Sub-Problem (SP) is solved at each iteration and a near-optimal set of column candidates G is returned. The greedy approach selects columns with the most negative reduced cost whereas the RL approach selects columns from G according to the Q-function learned by RLCG. RLCG converges faster and generates fewer iterations for the Cutting Stock Problem (CSP) and the Vehicle Routing Problem with Time Windows (VRPTW), compared to a greedy policy. In some cases it has similar performance to MILPs, although as the problem size increases, appears to outperform them as well.

**Questions:**

Column Generation is only introduced on the 3rd page. The authors need to restructure this as many readers will probably not be familiar with CG and will therefore not understand much in the first 2 pages.

You mention that RLCG is unable to train on instances of n=750 as it is expensive to solve such instances during training. Is scalability a fundamental limitation of the approach?

How sensitive is the curriculum training process? Is it easy to build a curriculum? What happens if you do not use a curriculum?

For the Vehicle Routing problem, how does RL compare to MILPs?

Line 39: The authors don’t seem to tackle the integer case which is what they claimed CG is mainly used for. As such, I’d like the authors to help me better understand the significance of these results.


**Limitations:**

The authors mention some limitations, such as adding a single column per CG iteration. However, there are a couple of other limitations that I think should be noted:

1.Scalability - it seems that, as the problem size increases, it becomes increasingly expensive to train the agent, to the point where it is not feasible. I do not have good intuition as to whether n=750 is difficult in the CSP problem for example, so it would be good for the authors to discuss the scalability challenges and significance of their results in more depth.

2. Performance compared to MILP - the authors need to be more specific as to how their method compares to MILP as they are both competitive. A statistical analysis would be important here to better understand where each method has its strengths and weaknesses. Replacing Figure 3 with this analysis would be helpful, as I dont get much insight from these figures.

3. When introducing the CG problem, the authors mention that it is the workhorse for tackling large scale integer linear programs. However, the authors then state that they do not tackle the integer case (line 39). This paper would be greatly strengthened, in my opinion, if the authors explain the significance of their results and/or provide an example tackling the integer case. A strong argument here will influence my score.


**Strengths And Weaknesses:**

Originality:
This paper appears to be original as I am unaware of RL being used for Column Generation. Utilizing graphnets to encode the state of the CG algorithm as well as the structure of the LP instance also appears to be a novel contribution.

Quality:
The paper is well written and the authors appear to do a good experimental comparison with the baseline algorithms. In addition, experimental results are shown on multiple domains which further strengthens the results.

Clarity:
Column Generation is only introduced on the 3rd page. The authors need to restructure this as many readers will probably not be familiar with CG and will therefore not understand much in the first 2 pages.

Figure 2: The axes are very hard to read

Significance:
It is difficult for me to judge the significance of the results. When introducing the CG problem, the authors mention that it is the workhorse for tackling large scale integer linear programs. However, the authors then state that they do not tackle the integer case (line 39). This paper would be greatly strengthened, in my opinion, if the authors explain the significance of their results or alternatively, provide an example tackling the integer case.

---

> ### Author Response · Authors · 2022-08-02
> **Comments to Reviewer 8Csh**
>
> Thank you very much for your careful review and constructive comments, which have much contributed to improving the paper. We will address all the comments in the paper and provide responses to the comments below.
>
> ### Scalability:
> Thank you for this helpful comment. We agree it can be confusing. We chose to not use samples with n=750 in training RLCG so we can show the upward generalization ability of our approach: we train on easy instances n=50, n=200, and we show that the learned policy can still perform well on hard instances n=750. The instances with n=750 are considered challenging tasks in the literature where the maximum problem size is n=1000. We can train RLCG using samples with n=750 or n=1000 and there is no scalability issue preventing us from doing so. We updated the discussion related to this section to make it more clear.
>
> ### Curriculum learning:
> We build the curriculum following the human intuition of the problem's difficulties starting with easier problems. The difficulty of instances is well studied for the datasets we chose. We built our curriculum using difficulty classifications from the references where we obtained our datasets. Appendix A shows the benefits of curriculum training. We are elaborating on this section and moving it into the main paper. Following Figure 7, the average number of iterations in 7a) is 60.79 and in 7b) is 70.44. We also compared the performance with curriculum learning and without curriculum learning on the CSP dataset using n=200 with 80 testing instances. We used the same configuration of hyperparameters (model 3 in Table 3) to run both experiments. For the curriculum learning the average number of iterations is 64.1 (std 9.6) and without the curriculum learning the average number of iterations is  75.5 (std 12) indicating a speed-up advantage by 15.1% when we use curriculum learning over testing instances. Across 80 instances, 91.25% of the time the curriculum learning model converged faster.
>
> ### RLCG vs MILP comparison for VRPTW:
> We have added new experiments benchmarking RLCG against the MILP expert using the VRPTW dataset. Our results confirm that RLCG consistently outperforms the MILP expert, which is the best one can achieve in one step look-ahead. We also compared RLCG with the expert on the VRPTW dataset with large instances including 20 large instances. On average, RLCG converged in 91.9 iterations while the expert converged in 117.8 iterations. RLCG outperforms the expert by 21.9% in terms of number of iterations. Among all the test instances, 85% of the time RLCG performs better. These results confirm again that RLCG consistently outperforms the expert in both datasets.
>
> ### The integer case:
> We addressed this question in the conclusions section in page 9 line 324 but will aim to clarify it earlier on in the paper. The CG algorithm solves the linear programming relaxation of an integer program with an exponential number of columns. Typically, the optimal solution to said LP relaxation may have some fractional variables. To obtain an integer-optimal solution, the well-known “branch and price” algorithm is used [add citation]. In a nutshell, it is a complete tree search algorithm which solves an LP relaxation with some additional integrality constraints at each node of the search tree. The CG algorithm is used to solve all those LP relaxations, making it the workhorse of MILP solving with a large number of variables. A trained RLCG agent can also be embedded within the branch and price algorithm for solving the integer-constrained versions of CSP/VRPTW and invoked to solve each LP relaxation at each node of the search tree. The speed-ups demonstrated in this paper would transfer to that setting: an average reduction of 20% in (per-node) LP running time from using a trained RLCG agent would, for example, result in a 20% reduction in MILP running time. Integrating an RLCG agent into existing branch-and-price solvers [reference some solvers, like VRPSolver] would require some software engineering as these solvers are C/C++-based whereas our implementation is in Python. We leave this exercise for follow-up work that can build on the open-source code we will make available.

---

### Official Review · Reviewer_Eksw · 2022-07-08

**Rating:** 5
**Confidence:** 4
**Soundness:** 2 fair
**Presentation:** 3 good
**Contribution:** 2 fair

**Summary:**

This work considers the problem of using machine learning (ML) to speed up the time to solve linear programmes (LPs) with a large number of columns (variables). Approaches to solving such problems fall under the category of column generation (CG); an approach whereby columns (variables) in the LP are iteratively selected from among the candidates to add to the restricted master problem (RMP) until no more columns with a negative reduced cost in the RMP are available, at which point a provably optimal solution to the LP will have been found. Classical approaches to CG typically follow a greedy policy whereby the column which leads to the most negative reduced cost is selected at each iteration. The authors posit that such a myopic approach does not necessarily lead to the fewest overall iterations to find the optimal solution to the LP. Instead, the authors propose a reinforcement learning for column generation (RLCG) method which treats column generation (CG) as a Markov decision process (MDP). Using this formulation, the authors train a policy parameterised by a graph neural network (GNN) with deep Q-learning (DQN) to, at each CG iteration, select the column which maximises the long-term return (which, in their setting, corresponds to minimising the overall number of CG iterations). The authors test their approach on the canonical cutting stock problem (CSP) and vehicle routing problem with time windows (VRPTW). They demonstrate that their method can outperform both a greedy heuristic and a 1-step-lookahead expert in terms of both solving time and decision quality (as measured by the overall number of CG iterations).

**Questions:**

* **Missing related work and context:** The authors discuss the work of Morabit et al. 2021, which they claim is the only prior work to have applied ML to CG. However, a quick search revealed there are at least a couple of other ML-for-CO approaches which have been proposed (e.g. Shen et al. 2021 and Babaki et al. 2022). The related work section should be more comprehensive to put the authors’ work in the context of the current literature.

* **Missing experimental baseline comparisons:**

    * Despite the availability of the above ML-for-CG algorithms, and despite the authors referring to the work of Morabit et al. 2021, the authors provide no comparison to any ML-for-CG baseline in the experiments section. Instead, the authors only compare to a simple greedy heuristic and to the 1-step-lookahead expert used by Morabit et al. 2021 for the data labeling process. The authors justify omission of the Morabit et al. 2021 ML agent by claiming that the expert they used provides an upper-bound on the best possible performance of the ML agent. While this is true in terms of decision quality, the ML agent of Morabit et al. 2021 should have a much faster solving time than that of the the 1-step-lookahead expert it imitates. Therefore, the authors should compare the solving time of RLCG to that of the ML agent of Morabit et al. 2021 - this should also enable more baselines than just greedy to be added to the VRPTW benchmark, which the authors claim is too computationally expensive to apply the 1-step-lookahead expert.

    * In relation to the above, the authors might also consider comparing RLCG to the work of Shen et al. 2021 and Babaki et al. 2022. In particular, Shen et al. 2021 appear to compare their ML-for-CG agent to a more comprehensive set of CG solvers than those considered by the authors (e.g. see Fig 2 of their paper). Are such baselines not relevant to the authors’ RLCG work?

* **Missing discussion of agent performance trend:** Why do the authors think in Fig 2 that the RL agent performs better than the expert in terms of number of iterations on the larger instances but not on the smaller instances (e.g. Fig 2a vs. Fig 2c)? This seems counter intuitive, since the expert is doing a 1-step lookahead regardless of the size of the problem, so I would have thought that the larger state-action space would have no effect on the expert but would present a familiar challenge for the RL agent. Is it really the case that this is down to longer episodes and therefore greater opportunity for non-myopic CG agents? Fig 2a already has episodes $40$-$80$ iterations long, which seems ample opportunity to outperform a 1-step lookahead policy. Would this indicate that, for some reason, non-myopic policies only have an advantage (in terms of decision quality) over myopic agents when making column selection decisions which can influence future selections $\geq 200$ steps later?

* **Curriculum learning analysis and discussion:** It is interesting that the authors emphasise the importance of curriculum learning on the final attainable results of RLCG. However, I do not fully follow the curriculum learning schedule or its effects described in the Appendix.
    *  In Fig 7 of the Appendix, the authors label their figures as ‘learning curves’. However, this is confusing terminology; usually learning curves show how the agent performs as a function of the number of training iterations/epochs it has undergone. The authors have instead plotted the instance index on the x-axis. As such, this is not really a curve showing how the agent is learning during training, but rather a plot of the agent’s performance for each instance. Furthermore, it is not clear what Fig 7 is showing; when indexing the instances from easy to hard and then randomly indexing the instances, is it not obvious that the former will lead to an upwards trend and the latter will lead to a random noise graph? How does this show the influence of curriculum learning on the agent in terms of learning stability and/or final attainable performance? I think a simple validation plot showing the performance of the curriculum and w/o curriculum agents on the validation set as a function of learner epochs would be more useful and demonstrate the efficacy of curriculum learning.
    * In Table 5 of the Appendix, the authors show how they split the instances into easy, medium, and hard instance types, but I cannot see what the curriculum schedule is in terms of when the agent progresses to the next level of difficulty. When does the agent move from easy to normal to hard? This should be carefully and clearly described given the claimed importance of curriculum learning.
    * Finally, although not essential, if space allows I would consider moving some of the curriculum learning results and/or their analysis and discussion into the main paper.

* **Analysis and discussion of proposed method's sensitivity to the introduced hyperparameter:** Does the sensitivity of RLCG to $\alpha$ in the reward function change for different problem types? What is the advantage of having this explicit balancing between the two components of the reward function?


### Miscellaneous minor issues

* Pg 7 Fig 2: The figure text is too small to read.

* Appendix pg 16 line 523 typo: ‘Similarly,the...’


### References

* Mouad Morabit, Guy Desaulniers, and Andrea Lodi. Machine-learning–based column selection for column generation. Transportation Science, 55(4):815–831, 2021.

* Yunzhuang Shen, Yuan Sun, Xiaodong Li, Andrew Eberhard and Andreas Ernst. Enhancing Column Generation by a Machine-Learning-Based Pricing Heuristic for Graph Coloring. AAAI, 2022.

* Behrouz Babaki, Sanjay Dominik Jena and Laurent Charlin. Neural Column Generation for Capacitated Vehicle Routing. AAAI ML4OR Workshop, 2022.

**Strengths And Weaknesses:**

Strong points:
* Considers an important application area of machine learning; improved column generation algorithms have the potential to address a variety of real-world continuous and discrete optimisation problems.
* For the most part, the paper is excellently written and easy to follow.
* The RL agent outperforms the brute-force 1-step-lookahead expert in terms of decision quality on larger instances, which is an impressive result.

Weak points:
* The related work is missing some key ML-for-CG references.
* The experiments section lacks a comparison to any ML-for-CG agent despite multiple options being available in the literature.
* The curriculum learning approach and results seem unclear.

---

> ### Author Response · Authors · 2022-08-02
> **Comments to Reviewer Eksw (1)**
>
> Thank you very much for your careful review and constructive comments, which have much contributed to improving the paper. We will address all the comments in the paper and provide responses to the comments below.
>
> ### Missing related work and context:
> Thank you for bringing important references to our attention. We added a paragraph in our literature review about Shen et al. 2021 and Babaki et al. 2022 (which appeared very recently online in a workshop which we were not aware of at the time of writing). These works are different from our current work in a sense that they are both doing static optimization as opposed to sequential decision-making. Shen et al. 2021 employ a ML model to predict which columns belong to the optimal basis. Babaki et al. 2022 are taking an imitation learning approach to mimic an expert that selects the column based on optimal duals similar to Morabit et al. 2021. Neither of these tackles the CG problem from a RL/sequential decision making viewpoint. Additionally, they each tackle a single problem class, namely Graph Coloring in Shen et al. and vehicle routing in Babaki et al., whereas we tackle two fairly distinct problem classes.
>
> ### Missing experimental baseline comparisons:
> Given one column selection, the one step look-ahead expert, which selects the candidate column that can bring down the objective the most, acts as an upper bound to any static heuristic including any ML methods. The related work listed here deals with static optimization instead of sequential optimization (including Babaki’s imitation learning approach). We also compared RLCG with the expert on the VRPTW dataset with large instances including 20 large instances. On average, RLCG converged in 91.9 iterations while the expert converged in 117.8 iterations. RLCG outperforms the expert by 21.9% in terms of number of iterations. Among all the test instances, 85% of the time RLCG performs better. Our results confirm again that RLCG outperforms the expert in both datasets in terms of number of iterations and solving time.
>
> ### Missing discussion of agent performance trend:
> The fact that the improvement of RLCG over the CG expert on harder instances is larger than improvement on simpler instances makes sense: the harder the instance, the longer the episode, the more advantage for taking the impact of future actions into consideration. For the smaller problems, the one step look-ahead performs well and there is less room for improvement. Even in cases where the numbers of iterations are similar, the RLCG is faster than the expert and the greedy algorithms (Figures 2d – 2f). This shows the benefit of formulating CG as a sequential decision making problem.
>
> ### Curriculum learning analysis and discussion:
> Both figures 7a) and 7b) show the agent’s performance in terms of number of iterations and both the x-axes show the instances the agent encounters during training (instances up to 200) and validation (instances 201 to 400). During training, in a) the agent encounters instances following the difficulty order, while in b) the agent encounters randomly ordered instances. It can be seen from the plots that in a) while there is an upward trend due to increase of instances' difficulty, within each difficulty level, there is a downward trend; while in b) there is no downward trend at all. Besides, as the y-axis for figures 7a) and 7b) have the same range, we can see that 7a) generally has a lower number of iterations than 7b) overall. The average number of iterations in 7a) is 60.79 and in 7b) is 70.44. The curriculum training follows strictly the instances order shown in Table 5 from top to the bottom (so 160 easy instances -> 160 medium instances -> 80 hard instances). We have added more details to explain how we used the training samples in the curriculum learning. We will also move the curriculum learning discussion into the main paper as suggested.
>
> We also compared the performance with curriculum learning and without curriculum learning on the CSP dataset using n=200 with 80 testing instances. We used the same configuration of hyperparameters (model 3 in Table 3) to run both experiments. For the curriculum learning the average number of iterations is 64.1 (std 9.6) and without the curriculum learning the average number of iterations is  75.5 (std 12) indicating a speed-up advantage by 15.1% when we use curriculum learning over testing instances. Across 80 instances, 91.25% of the time the curriculum learning model converged faster.

---

> > ### Author Response · Authors · 2022-08-02
> > **Comments to Reviewer Eksw (2)**
> >
> > ### Analysis and discussion of proposed method's sensitivity to the introduced hyperparameter:
> > The alpha parameter serves as a scale factor for the importance of reducing the objective value compared to minimizing the number of iterations. To avoid redoing an expensive hyperparameter search for VRPTW, we opted to use the best hyperparameter configuration found for CSP on VRPTW. As the results were satisfactory, we did not find the need to perform a sensitivity analysis of alpha or a separate hyperparameter search for VRPTW. However, such a search could be performed and would only improve our final results, if a better value for alpha is found. It is worth noting that Table 3 of the appendix shows that whenever alpha=0, i.e. the reward is -1 for each additional CG iteration without any consideration for the bound improvement, the validation reward is low. Whenever alpha is large, better results are obtained.
> >
> > ### Miscellaneous minor issues:
> > We have made the corrections. Thank you very much.

---

> > > ### Comment · Reviewer_Eksw · 2022-08-07
> > > **Reviewer Eksw response**
> > >
> > > I thank the authors for taking the time to address my points.
> > >
> > > - **Missing experimental baseline comparisons**:
> > >
> > >     - **Morabit et al. 2021**: I think the authors have misunderstood my point here. I understand that the one-step-lookahead expert provides an upper bound on the best possible performance of the so-called 'static' optimisation methods in terms of *decision quality*. However, it does not give an upper bound on the best performance of methods which imitate one-step-lookahead experts in terms of *solving time*. This is because a neural network imitating an expert will have lower inference times than the original expert, and therefore should have faster solving times. Therefore, the authors should compare their work to the Morabit et al. 2021 baseline in terms of *solving time*.
> > >
> > >     - **Shen et al. 2021**: The authors have not addressed my point about the far more comprehensive set of CG baselines shown by Shen et al. 2021, which I think should also be included in the authors' paper.
> > >
> > > - **Missing discussion of agent performance trend**: Here the authors have not directly addressed my question, they have just re-stated what is already in the paper. I understand that the authors claim that longer episodes provide greater opportunity for non-myopic CG agents, and that this is why RL outperforms on these problems. But as I stated in my review, Fig 2a already has episodes $40$-$80$ iterations long, which seems ample opportunity to outperform a 1-step lookahead policy with a non-myopic policy. Why are $40$-$80$ episode steps not enough to outperform one-step-lookahead policies? Why do non-myopic policies only have an advantage (in terms of decision quality) over one-step-lookahead agents when making column selection decisions which can influence future selections $200$ steps later?
> > >
> > > - **Curriculum learning**: Can the authors not provide an additional standard learning curve validation plot with average performance on the validation set on the y-axis and number of epochs on the x-axis to compare the curriculum and non-curriculum learning agents?

---

> > > > ### Author Response · Authors · 2022-08-08
> > > > **Part 1 of response**
> > > >
> > > > Thank you for engaging with our rebuttal, we do appreciate all the clarifications you’ve made and hope that the following response will be convincing.
> > > >
> > > > * __Missing experimental baseline comparisons:__
> > > >   * __Morabit et al. 2021:__ Thank you for clarifying; we initially misunderstood your point here. We would like to justify our choice of not comparing with Morabit here with two set of inequalities / equalities:
> > > >
> > > >     (1) Time per iteration (RLCG) = Time per iteration (Morabit) << Time per iteration (Expert)
> > > >
> > > >     The “=” is due to the fact that our GNN architecture and number of features are comparable to Morabit’s supervised learning model. The “<<” is due to the Expert doing expensive computations while the Morabit ML agent is imitating the expert; similarly, RLCG is much faster than the Expert per iteration.
> > > >
> > > >     (2) Number of CG steps (RLCG) < Number of CG steps (Expert) < Number of CG steps (Morabit)
> > > >
> > > >     The first “<” is demonstrated in our experiments; the second “<” is because the Morabit model is imitating the Expert, and is thus bound to perform worse than the Expert due to errors in predictions on unseen data.
> > > >
> > > >     Since the Total running time = num iterations*number of CG steps, and given these two sets of inequalities, we have that:
> > > >
> > > >     Total running time (RLCG) < Total running time (Morabit), due to having a smaller number of iterations and a comparable time per iteration. As such, we find that there is no need for a direct comparison with Morabit.
> > > >
> > > >   * __Shen et al. 2021:__ We are grateful that you brought up this point and we will make the following clarification: the baselines in their paper consist of different pricing algorithms rather than column selection strategies. In CG, the pricer (which is used to generate candidate columns) can be independent of the selection method (which selects  out of already generated candidate columns). Shen et al. tried to replace the pricer. In our case, the pricing is done by exactly solving the subproblem in Gurobi and this pricing is used across all column selection strategies (Expert, Greedy and RLCG).  In short, they don’t have a CG baseline, they only have pricing baselines. Such baselines do not apply to our case.
> > > >
> > > >     Therefore, we believe that by comparing RLCG to greedy selection and Expert selection, we cover a spectrum of different selection methods from very myopic greedy to the one step expensive look-ahead expert. We would have liked to compare with other baselines that reason about the long-run sequential nature of the CG process (which RLCG does), but we are not aware of any work that considers this.
> > > >
> > > >
> > > > * __Missing discussion of agent performance trend:__ We apologize for our initial answer to this point; we hadn’t fully grasped your argument.
> > > >
> > > >     First of all,  it is *not* the case that RLCG is not outperforming the Expert even on the small instances of Fig. 2a. In Figure 2a), for 59.2 % of the testing instances, RLCG converged in fewer iterations than the Expert (the points above the  diagonal in the figure). In appendix Section J, Table 6, we also show that RLCG converges faster than the Expert on average on small instances (41.6 steps vs 42.5 steps). While small a difference, RLCG is indeed outperforming the Expert even on the smallest set of CSP instances.
> > > >
> > > >     Secondly, as to your point about why 40-80 iterations are not enough to demonstrate a big gap between RLCG and Expert, we hypothesize the following:
> > > >
> > > >     Due to those n=50 instances being relatively easy, both RLCG and the Expert are already close to “optimal” in terms of number of iterations. This would limit the room for improvement via learning. Unfortunately, it is difficult to validate this hypothesis because getting the optimal (least-number-of-iterations) selection policy is intractable.
> > > >
> > > >     We conclude this comment by noting that our results conclusively demonstrate that RLCG outperforms the Expert especially as instance size grows.

---

> > > > > ### Author Response · Authors · 2022-08-08
> > > > > **Part 2 of response**
> > > > >
> > > > > * __Curriculum learning:__ First, we would like to clarify the confusion (due to misleading terminology on our part) about Figure 7 being a learning curve, in particular the following comments in your original review:
> > > > > “In Fig 7 of the Appendix, the authors label their figures as ‘learning curves’. However, this is confusing terminology; usually learning curves show how the agent performs as a function of the number of training iterations/epochs it has undergone. The authors have instead plotted the instance index on the x-axis. As such, this is not really a curve showing how the agent is learning during training, but rather a plot of the agent’s performance for each instance.”
> > > > >
> > > > >     A learning curve plots the agents’ reward as training iterations proceed. Indeed, this is what Figure 7 plots are showing: each training iteration consists of the agent processing one instance (shown in the x-axis), and the corresponding number of steps till CG convergence is shown on the y-axis. We agree that the term “Instance index” is misleading and will adjust it to “Training iteration”.
> > > > >
> > > > >     Secondly, the curriculum learning agent outperformed the non-curriculum learning agent both in terms of average number of steps on the training instances during training (60.79 steps vs 70.44 steps) and on the held-out validation set with n = 200 instances (64.1 steps vs 75.5 steps).
> > > > >
> > > > >     Lastly, when we trained the agent with or w/o curriculum, we did not track the validation set performance over the iterations. We only checked at the termination of training and concluded, based on the training and validation statistics mentioned in the previous paragraph, that curriculum learning gives a better policy. To generate another pair of plots showing validation instead of training performance during curriculum/non-curriculum training, we would need to re-run the whole training pipeline and solve all validation instances many times during the training process, which we are unable to do within the discussion time frame. However, we will perform this experiment for the final version of the paper, although we do not believe it will alter any of our conclusions.
> > > > >
> > > > >
> > > > >     Note: We apologize for a typo in our last response to section Curriculum learning analysis and discussion: we made a typo in “Both figures 7a) and 7b) show the agent’s performance in terms of number of iterations and both the x-axes show the instances the agent encounters during training (instances up to 200) and validation (instances 201 to 400)”. Please ignore/remove “(instances up to 200) and validation (instances 201 to 400)” as all instances here are training as clarified in these new comments we made.

---

### Official Review · Reviewer_DMVD · 2022-07-12

**Rating:** 7
**Confidence:** 4
**Soundness:** 4 excellent
**Presentation:** 4 excellent
**Contribution:** 4 excellent

**Summary:**

The paper proposes a reinforcement-learning-based approach for column selection within a column generation procedure to solve an LP relaxation. Candidate columns come from a set of solutions encountered while solving the column generation subproblem, and Q-learning is applied to choose a column, rewarding large changes in objective value and few iterations. Computational results on cutting stock and vehicle routing problems show that this approach reduces number of iterations and solve times by a significant margin compared to standard column generation (at the cost of training on instances from the same family).

**Questions:**

Minor suggestions related to clarity and details:

* Please specify in Appendix C which PoolSearchMode parameter you are using in Gurobi to find the 10 candidate columns. It is useful to know if these are the top 10 candidates, or 10 incidental candidates during search.

* In Appendix C, it refers to the code for the network structure. Could you please describe the network structure in the appendix?

* Notes on text: Missing figure reference in line 176. Typo "hyperparemeters" in line 304. Some of the citations need to be with parenthesis.

I have a few questions that are not necessary for this submission, but mostly for my own curiosity:

1. Have you considered keeping some of the older column candidates, perhaps with some system to age them out of the graph as needed? This could be helpful in two ways: a previous good column is still there if the RL agent ends up choosing a bad column by mistake, or if there is more than one column worth selecting in an iteration.

2. Have you considered adding more than one column per iteration (i.e. between subproblem solves)? This would be an easy way to consider multiple columns. It might work better if you keep older candidates as described above.

3. Do you know what features were important to select good columns? If this is insightful (e.g. if a particular feature that is not reduced cost stands out, or if it is still mainly reduced cost), I believe this would be a nice contribution to include in the paper. It would be great to know if one can extract heuristic insights that can be incorporated back into standard column generation, without lengthy training times. In addition, I am curious how much the problem-specific features matter, since ideally it would be good to have this work out-of-the-box without problem-specific features.

**Limitations:**

No further limitations other than the ones discussed above. The paper does a decent job at discussing its limitations at the end.

**Strengths And Weaknesses:**

This paper is a solid step in improving the scalability of problems in Operations Research using machine learning techniques. Column generation is an important technique to solve large-scale problems with exponentially-many columns, and the paper provides concrete evidence that a reinforcement-learning-based approach can be successful at selecting good columns, assuming one has a set of instances that can be trained on. While the RL formulation and neural network architecture mostly follow the literature standard, it is sound and novel (as far as I know). Furthermore, it is useful to know that curriculum learning helps in this approach. The computational experiments are well-designed and reasonably extensive: it includes hyperparameter tuning and a good number of practical instances. The improvements from this method are substantial in the context of machine learning techniques for OR, particularly for larger instances.

I do not have any major concerns with the paper. The paper might have been slightly stronger if it compared with column generation with stabilization, but the method is already taking fewer iterations than a look-ahead expert, and thus this comparison might not be necessary.

Overall, everything looks solid in the paper and I believe this is a valuable contribution. I recommend this paper to be accepted.

---

> ### Author Response · Authors · 2022-08-02
> **Comments to Reviewer DMVD**
>
> Thank you very much for your careful review and constructive comments, which have much contributed to improving the paper. We will address all the comments in the paper and provide responses to the comments below.
>
> ### Minor suggestions:
> We have made the corrections. Thank you very much.
>
> ### 1. Column choice:
> We have not considered this approach but we agree it could be useful. We will think about good ageing-out rules that could enable such an idea.
>
> ### 2. Multiple columns:
> Indeed, we are considering this direction for future work. Modeling the Q-value for subsets of columns rather than a single column may be tricky, but the heuristic you suggest of adding columns from previous iterations is interesting, and we thank you for this idea.
>
> ### 3. Feature selection:
> We have not performed such a feature selection or feature importance analysis. The vast majority of the features we used are not problem-specific making out-of-the-box application of the framework minimally cumbersome. This is how we were able to tackle both CSP and VRPTW with essentially the same implementation.

---

### Public Comment · ~Cheng_Chi3 · 2023-02-23
**github link to our submission**

The hyperlink in the pdf is not working, below is the link to our github repo:
https://github.com/khalil-research/RLCG.git

---

### Meta-Review · Area_Chair_hcAq · 2022-09-05

**Recommendation:** Accept
**Confidence:** Certain

**Metareview:**

The reviewers all agree that the paper meets the acceptance bar.

At the same time, I would like to encourage the authors to seriously consider adding more experiments to the final paper as recommended by the reviews:
1. Given the motivation in the paper, the recommendation of Reviewer 8Csh about running some experiments for MIP is quite reasonable, and would make the story much more convincing. It would be a significant improvement, worth the extra software engineering efforts.
2. I do not quite agree with the reasons the authors rejected comparison with algorithms recommended by Reviewer Eksw. It was argued that these algorithms are superseded by the one-step lookahead algorithm. However, the latter is a greedy algorithm, while the ML based methods may deviate from it, which could be beneficial (especially, Babaki et al can try to learn a better sequential baseline if available). Some of these papers also consider adding multiple columns, as also suggested by Reviewer DMVD, and accepted as future work by the authors. Hence, comparison to these papers could partially answer that question.
3. Including the validation curves for curriculum learning as requested by Reviewer Eksw would also be quite interesting.

I sincerely hope that authors can run these experiments while preparing the final version.

**Award:**

No

---

### Decision · Program_Chairs · 2022-09-14

Accept